*Review Article*

# Interplay between fungal infections and autoimmunity: mechanisms and therapeutic perspectives

Devon T DiPalma [1], Miranda K Lumbreras [1] & Mari L Shinohara [1,2,3,4]✉

## Abstract

**Fungal infections and autoimmunity share a complex, bidirectional relationship that significantly impacts patient outcomes. Emerging evidence highlights how fungal pathogens contribute to autoimmune processes by triggering immune dysregulation. Conversely, autoimmunity and its immunomodulatory treatments increase susceptibility to fungal infections. These interactions manifest through altered immune responses, including changes in inflammatory signaling, antigen recognition, and mycobiome composition. The resulting interplay complicates disease management, necessitating careful balancing of antifungal defenses with immune modulation. This review synthesizes current knowledge on the role of fungal infections in the progression of autoimmune conditions and explores how autoimmune diseases predispose individuals to fungal infections. Key insights emphasize the need for integrative treatment approaches, addressing both infection risks and immune system imbalances.**

**Keywords** Fungal Infections; Autoimmunity; Mycobiome; Antifungal Therapeutics
**Subject Categories** Immunology; Microbiology, Virology & Host Pathogen Interaction

## Introduction

Fungi are ubiquitous microorganisms with significant roles in human health and disease. Although traditionally studied for their pathogenic capabilities, fungi are increasingly recognized for their involvement in autoimmune diseases. Fungal pathogens have evolved mechanisms to invade human tissues, evade immune responses, and persist within the host. Concurrently, these pathogens and the broader mycobiome—the fungal component of the microbiome—exert profound immunomodulatory effects that can predispose susceptible individuals to autoimmunity.

Emerging evidence links fungal pathogens to autoimmune diseases, suggesting possible mechanisms such as dysbiosis, molecular mimicry, and immune dysregulation. This review explores the complex interplay between fungal infections and autoimmunity, focusing on how immune responses to fungi contribute to disease progression and how antifungal treatments might influence these interactions. Advancing our understanding of fungal–host immune dynamics is crucial for developing targeted therapies and improving outcomes for patients with fungal infections and autoimmune diseases.

## Fungal pathogens and immune cells

This section outlines the fungal pathogens associated with autoimmune conditions. It will also discuss the roles of innate, innate-like, and adaptive immune cells in combating fungal infections, highlighting their overlap as key players in autoimmunity.

### Fungal pathogens associated with autoimmune conditions

Pathogenic fungi must meet the following criteria to successfully infect humans: the ability to thrive at or above 37 °C, penetrate biological barriers, degrade host tissue, and evade or withstand the immune system (Kohler et al, 2014; Kohler et al, 2017). Only a limited number of fungal species have evolved to fulfill these requirements. In October 2022, the World Health Organization (WHO) released the "Fungal Priority Pathogens List" (FPPL), highlighting the most critical fungal pathogens that pose a significant threat to public health and require urgent research and development (WHO, 2020). Among these pathogens, four species have been identified as top-priority concerns: *Aspergillus fumigatus*, *Cryptococcus neoformans*, *Candida albicans*, and *Candida auris*. Some of these species, along with others, have been cited in the literature to have an association with autoimmunity and immune-mediated diseases (Table 1).

*C. albicans* is a common opportunistic fungal pathogen but also thrives as a commensal organism in mucosal tissues (Gow et al, 2011; Talapko et al, 2021). Given its presence in the gut mycobiota, research linked *C. albicans* to autoimmune diseases such as multiple sclerosis (MS) (Benito-Leon and Laurence, 2017; Benito-

¹Department of Integrative Immunobiology, Duke University School of Medicine, Durham, NC, USA. ²Department of Molecular Genetics and Microbiology, Duke University School of Medicine, Durham, NC, USA. ³Department of Cell Biology, Duke University School of Medicine, Durham, NC, USA. ⁴Department of Neurobiology, Duke University School of Medicine, Durham, NC, USA. ✉E-mail: mari.shinohara@duke.edu

**Glossary**

| | |
|---|---|
| **Fungal priority pathogens list (FPPL)** | Published by the World Health Organization in October 2022, this list identifies fungal pathogens posing the greatest public health threats. The Critical Priority Group includes *Candida auris, Candida albicans, Aspergillus fumigatus*, and *Cryptococcus neoformans* based on the public health importance, determined by antifungal resistance and disease-burden-related criteria (mortality, annual incidence, and morbidity). |
| **Ascomycota** | The largest phylum of fungi, also known as sac fungi, characterized by the production of sexual spores called ascospores, which are formed inside specialized sac-like structures called asci. Ascomycota includes both unicellular organisms (e.g., *Saccharomyces*) and multicellular forms (e.g., *Aspergillus, Candida, Histoplasma, Coccidioides*). Their asexual reproduction typically involves conidia, which are formed on specialized hyphae. Medically important genera within this group, including *Candida* and *Aspergillus*, are implicated in superficial, mucosal, and invasive fungal infections, particularly in immunocompromised patients. Ascomycetes are also known for their antifungal resistance patterns, making accurate diagnosis and appropriate treatment essential in clinical settings. |
| **Basidiomycota** | A fungal phylum characterized by its production of sexual spores called basidiospores on structures called basidia. While many members of this phylum are non-pathogenic and include mushrooms, puffballs, and shelf fungi, several are of clinical significance. Notable pathogens include *Cryptococcus neoformans* and *C. gattii*, which cause meningitis in immunocompromised individuals. *Malassezia*, a genus of lipophilic yeasts, is found as part of the normal skin microbiota in humans and animals. Clinically, *Malassezia* species are associated with several superficial skin conditions |
| **Mononuclear phagocytes (MNPs)** | Myeloid cells with a single round nucleus that can engulf and digest pathogens, dead cells, and debris. This group includes monocytes, macrophages, and dendritic cells. They play key roles in innate immunity, inflammation, and antigen presentation, helping to bridge innate and adaptive immune responses. |
| **Mycobiome** | Refers to the community of fungi that inhabit various body sites as part of the broader human microbiome. Typical genera include *Candida, Malassezia, Aspergillus*, and *Cladosporium*, which are typically present on the skin, in the gut, respiratory tract, and other mucosal surfaces. The mycobiome interacts closely with the host immune system and bacterial microbiota, contributing to immune regulation, barrier integrity, and inflammation. Advances in sequencing technologies have enabled deeper insights into the composition and function of the human mycobiome, though it remains less well-characterized than its bacterial counterpart. |
| **Trained immunity (TI)** | Refers to the phenomenon in which innate immune cells, such as monocytes, macrophages, and natural killer (NK) cells, undergo epigenetic, transcriptional, and metabolic reprogramming after an initial exposure to a pathogen or stimulus. This reprogramming enhances their ability to respond more robustly to subsequent infections, even those caused by unrelated pathogens. Unlike classical adaptive immunity, which relies on antigen-specific T and B lymphocytes and memory formation, trained immunity is non-specific and does not involve gene rearrangement or clonality. It can be induced by microbial components (e.g., β-glucans, BCG vaccine) or endogenous signals (e.g., oxidized LDL), lasting from weeks to months. |
| **Molecular mimicry** | Phenomenon in which structural similarities between foreign antigens (such as from microbes) and host self-antigens lead the immune system to mistakenly target the body's own tissues. This misrecognition can result in autoimmune responses, as immune cells that are primed to attack the pathogen may also attack host tissues with shared or similar molecular features. |
| **Epitope spreading** | Immunological phenomenon in which the immune response initially targets specific epitopes (distinct regions of an antigen), but over time expands to recognize additional epitopes—either on the same antigen (intramolecular spreading) or on different antigens (intermolecular spreading) secondary to inflammatory tissue destruction. Clinically, epitope spreading is particularly relevant in autoimmune diseases. The immune system may begin by targeting a single self-antigen and later expand to attack multiple self-components, worsening the disease. This progression complicates treatment and is thought to contribute to chronicity and relapses. |
| **Dectin-1** | A CLR, encoded by the *CLEC7A* gene and expressed on innate immune cells such as dendritic cells, macrophages, neutrophils, and monocytes. Dectin-1 detects β-glucans, key components of fungal cell walls. Upon ligand binding, |

| | | | |
|---|---|---|---|
| | Dectin-1 activates intracellular signaling through the Syk–CARD9 pathway, leading to cytokine production, reactive oxygen species generation, phagocytosis, and promotion of antifungal immunity, including Th17 responses. Genetic variants in *CLEC7A* have been linked to increased susceptibility to fungal infections and potentially to autoimmune or inflammatory conditions. | Immunomodulatory therapies | LAG-3, TIM-3) that act as brakes on T-cell activity. By inhibiting these checkpoints, ICIs release the suppression of T cells, enabling a more robust immune attack on tumor cells. Treatments designed to modify the immune system's activity to achieve a desired therapeutic effect. These therapies can either enhance or suppress immune responses to reduce harmful inflammation, as seen in autoimmune diseases or transplant rejection. They encompass a wide range of agents, including monoclonal antibodies, small molecules, cytokines, immune checkpoint inhibitors, and biologics targeting specific immune pathways. |
| β-glucans | Polysaccharides composed of D-glucose units linked primarily through β-(1 → 3)- and/or β-(1 → 6)-glycosidic bonds, commonly found in the cell walls of fungi, yeast, certain bacteria, algae, and cereals like oats and barley. In the context of fungi, β-glucans are structural components that play a crucial role in maintaining cell wall integrity and are recognized by host immune cells through PRRs such as dectin-1. | Anti-Saccharomyces cerevisiae antibodies (ASCAs) | Serologic markers directed against mannan components of the cell wall of *S. cerevisiae*. Although *S. cerevisiae* is generally considered non-pathogenic, ASCAs are frequently detected in patients with Crohn's disease and other immune-mediated disorders, serving as a potential indicator of abnormal immune responses to commensal or environmental fungi. |
| Immune checkpoint inhibitors (ICIs) | A class of drugs that enhance the body's immune response against cancer by blocking regulatory pathways that normally restrain immune activation. These therapies target checkpoint proteins (e.g., PD-1/PD-L1, CTLA-4, | | |

León et al, 2010; Yadav et al, 2022), type-1 diabetes (T1D) (Gursoy et al, 2018; Honkanen et al, 2020), and rheumatoid arthritis (RA) (Lee et al, 2022). Other *Candida* species, such as *C. glabrata* and *C. krusei*, have also been implicated in autoimmunity (Fraga-Silva et al, 2022). Their association with autoimmune disorders may stem from their prevalence in the human body and high infection rate, especially in females (Denning et al, 2018).

The *Cryptococcus* genus includes two major species that infect humans: *C. neoformans* (including recently subcategorized *C. deneoformans*) and *C. gattii*. *C. neoformans* primarily infects immunocompromised individuals, whereas *C. gattii* can infect healthy individuals (Kwon-Chung et al, 2014; Li and Mody, 2010). These encapsulated fungi pose a serious threat to patients with T-cell deficiencies, such as AIDS patients, where cryptococcal infection often progresses to meningoencephalitis (Brizendine et al, 2013). *Cryptococcus* infection has also been linked to MS immunomodulatory therapies, which will be discussed in the section "Studies highlighting autoimmune diseases and fungal infections".

Among the *Aspergillus* species, *A. fumigatus* is a major human fungal pathogen. This saprophytic fungus generates airborne conidia, which can cause pulmonary infection in immunocompromised individuals. Studies have connected *Aspergillus* species with autoimmune diseases, such as MS (Shah et al, 2021) and Systemic Lupus Erythematosus (SLE) (Yang et al, 2023), in addition to the immune-mediated, but not autoimmune, Crohn's disease (CD) (Li et al, 2014).

## Myeloid cells

The host immune system adapts to fungal infections, balancing protection and potential autoimmune activation (Table 2). Myeloid cells are key antifungal defenders, acting as pathogen detectors, phagocytes, and antigen-presenting cells (APCs). They recognize

fungi mainly through pattern recognition receptors (PRRs), including but not limited to Toll-like receptors (TLRs) and C-type lectin receptors (CLRs). Among CLRs, Dectin receptors (Dectin-1, Dectin-2, and Dectin-3/MCL) and Mincle are particularly well-studied because they detect key fungal components (Shiokawa et al, 2017), such as β-glucan, α-mannan, and glycolipid, and facilitate the recruitment and activation of spleen tyrosine kinase (Syk). This process activates NF-κB through the CARD9–BCL10–MALT1 signaling complex. The subsequent release of proinflammatory cytokines and chemokines, including interleukin-23 (IL-23), interleukin-1β (IL-1β), tumor necrosis factor-alpha (TNFα), and CXCL1, promotes T helper 17 (Th17) cell responses (LeibundGut-Landmann et al, 2007).

Neutrophils are recruited early in infections, clear fungi through phagocytosis, generate reactive oxygen species (ROS), produce antifungal molecules, and increase production of proinflammatory molecules, as well as neutrophil extracellular traps (NETs) formation (Desai and Lionakis, 2018; Urban and Backman, 2020). Neutropenia increases susceptibility to invasive fungal infections (IFIs) (Herbrecht et al, 2000). Recent studies indicated that subpopulations of neutrophils with suggested distinct functions emerged during fungal infection (Deerhake et al, 2021b).

Macrophages can be tissue-resident or monocyte-derived. Tissue-resident macrophages are generally long-lived and act as immune sentinels (Kanayama et al, 2015; Xu and Shinohara, 2017; Xu-Vanpala et al, 2020). Monocyte-derived macrophages are crucial to clear fungal infections (Kanayama et al, 2015; Ngo et al, 2014), but they can also act as fungal reservoirs (Gilbert et al, 2014; Heung, 2020). Unresolved inflammatory responses by macrophages may contribute to autoimmunity.

Dendritic cells (DCs), the main APCs, link innate and adaptive immunity. They initiate antifungal Th1 and Th17 responses, as well

**Table 1. Fungal species associated with autoimmune and inflammatory diseases.**

| Fungal genus/species | Autoimmune disease | Inflammatory disease | Proposed roles | References |
|---|---|---|---|---|
| *Candida albicans* | MS, RA, T1D | CD, UC | Th17 stimulation; gut dysbiosis; translocation; immune priming | Sokol et al, 2017; Benito-León and Laurence, 2017; Benito-León et al, 2010; Yadav et al, 2022; Gursoy et al, 2018; Honkanen et al, 2020; Lee et al, 2022; Li et al, 2014 |
| *Candida tropicalis* | | CD | Gut enrichment; interkingdom interactions | Hoarau et al, 2016 |
| *Candida krusei* | MS | | Enhancing Th1/Th17 responses | Fraga-Silva et al, 2022 |
| *Candida galabrata* | MS; SLE | CD | Enhancing Th1/Th17 responses; | Liguori et al, 2016; Fraga-Silva et al, 2022; Yang et al, 2023 |
| *Saccharomyces cerevisiae* | T1D, SLE, MS, RA | CD | Target of ASCA antibodies; loss of tolerance | Main et al, 1988; Sendid et al 2024; Sokol et al, 2017 |
| *Aspergillus* spp. | MS | CD, ABPA | Th2/Th17 stimulator; hypersensitivity | Li et al, 2014; Shah et al, 2021;Chauhan et al, 2000 |
| *Cryptococcus neoformans* | MS | CD | CNS tolerance disruption, Immunomodulatory therapies | Li et al, 2014; Scotto et al, 2021; Nasir et al, 2023 |
| *Malassezia* spp. | SLE | CD | Drives inflammation via CARD9 | Limon et al, 2019; Yang et al, 2023 |

as control the proliferation of regulatory T cells (Tregs) (Yamazaki et al, 2003). Although DCs activate T cells against fungi, they may also initiate autoimmunity by presenting autoantigens (Saferding and Bluml, 2020).

## Innate lymphocytes

Innate lymphocytes include natural killer (NK) cells, natural killer T (NKT) cells, innate lymphoid cells (ILCs), γδT cells, and mucosal-associated invariant T (MAIT) cells. NK cells protect against fungi through direct killing with perforin and granulysin, and serve as a key source of interferon-gamma (IFNγ) (Li et al, 2013; Ma et al, 2004; Park et al, 2009). ILCs in the oral mucosa produce IL-17 during oropharyngeal candidiasis (Gladiator et al, 2013), aiding neutrophil recruitment. γδT cells contribute to defense by producing cytokines like IL-17 (Akitsu and Iwakura, 2018; Monin et al, 2020) and target *C. albicans* (Fenoglio et al, 2009; Maher et al, 2015; Poggi et al, 2009), *A. fumigatus* (Hebart et al, 2002), and *Paracoccidioides brasiliensis* (Munk et al, 1995). γδT cells are critical in mucosal immunity against *C. albicans* (Monin et al, 2020). Although less studied, MAIT cells are activated by conidia of *Aspergillus* spp. on APCs in a T-cell receptor (TCR)-dependent manner (Jahreis et al, 2018).

## T and B cells

The adaptive immune system also plays a critical role in fungal defense, as evidenced by increased susceptibility in immunocompromised patients (Brown et al, 2012; Wuthrich et al, 2012). The involvement of CD4[+] T cells is well-studied in antifungal immunity and antifungal vaccine development, with growing research on CD8[+] T-cell contributions. Th17 responses maintain epithelial barriers and recruit neutrophils, whereas Th1 responses sustain IFNγ production (Hernandez-Santos and Gaffen, 2012). Th1 and Th17 are protective, but Th2 responses can be detrimental to fungal infections (Wuthrich et al, 2012). Th1 and Th17 cell responses also contribute to autoimmune diseases such as MS and RA. In contrast, Tregs negatively control and balance immune responses, though a study suggested Tregs could promote Th17 differentiation and enhance host resistance in a candidiasis model (Pandiyan et al, 2011). Increased Th17/Treg ratios are linked to autoimmune diseases (Lee, 2018), highlighting the need for balanced Th cell responses. CD8[+] T cells have been less studied, but studies have suggested their protective role in general, particularly in histoplasmosis (Deepe, 1994; Nanjappa et al, 2012a; Nanjappa et al, 2012b).

B cells contribute to antifungal immunity via antibody production, cytokine production, and antigen presentation. They generate antibodies against fungal polysaccharides, such as β-glucans and glucuronoxylomannan (GXM) in *C. neoformans* (Casadevall and Pirofski, 2012). Antibodies exert fungicidal effects through methods of killing, opsonization, complement activation, and antibody-dependent cellular toxicity (ADCC) with evidence from *C. albicans*, *C. neoformans*, and *Histoplasma capsulatum* (Han et al, 2001; Nabavi and Murphy, 1986; Shi et al, 2008). However, excessive B-cell activity, ectopic germinal centers, proinflammatory cytokine production, and autoantibody production can promote autoimmunity. It is possible that memory B cells trigger autoimmunity if fungal antigens cross-react with self-epitopes.

Table 2. Representative bidirectional roles of major leukocyte types in fungal infection and autoimmunity.

| | Fungal infections | | Autoimmune diseases | |
|---|---|---|---|---|
| | Beneficial to host | Detrimental to host | Beneficial to host | Detrimental to host |
| **Neutrophils** | Phagocytosis and ROS generation<br>Secretion of antifungal molecules<br>NETs to trap or kill fungi<br>Production of proinflammatory cytokines and chemokines | Excessive inflammation causing tissue damage | Regulating immune responses (e.g., granulocytic MDSC)<br>Resolution of inflammation by secreting anti-inflammatory mediators | Autoantigen exposure by NETosis<br>Promoting Th17 responses |
| **Macrophages/ monocytes** | Fungal recognition through Pattern Recognition Receptors (PRRs)<br>Fungal clearance by phagocytosis<br>Pathogen containment by granuloma formation<br>Antigen presentation | Reservoir for fungal survival<br>Excessive inflammation causing tissue damage<br>Chronic inflammation or fibrosis | Tissue repair (M2 phenotype)<br>Clearance of apoptotic cells and debris<br>Regulating immune responses (e.g., monocytic MDSC) | Excessive inflammation and tissue damage (M1 phenotype)<br>Chronic inflammation and fibrosis<br>Proinflammatory cytokine production |
| **Dendritic cells** | Fungal antigen presentation and initiation of antifungal Th1/17 responses<br>Fungal recognition through PRRs<br>Pathogen containment by granuloma formation | Promote regulatory phenotypes of T cells<br>Overactivation leads to immunopathology and tissue damage | Initiating T-cell tolerance | Production of autoantibodies<br>Antigen presentation<br>Proinflammatory cytokine production<br>Formation of ectopic germinal centers (EGCs) |
| **T cells** | Antifungal activities by Th1, Th17, and CD8+T cells<br>Th17 responses to maintain epithelial barriers and neutrophil recruitment<br>Long-term immunity | Lung pathology by Th2 cells<br>Immune suppression by Tregs<br>Excessive inflammation and tissue damage by Th1 and Th17 | Tregs to control autoimmunity<br>Production of anti-inflammatory cytokines<br>Reestablishment of peripheral tolerance | Autoreactive T cells<br>Proinflammatory Th1 and Th17 cells<br>Cytotoxic T cells |
| **B cells** | Antibody production for opsonization, neutralization of fungal toxins, and complement system activation<br>Cytokine production<br>Antigen presentation | Regulatory responses (e.g., IL-10-producing B cells)<br>Overproduction of IgE | Bregs to control autoimmunity<br>Autoantibodies to clear apoptotic debris and cellular waste<br>Promote anergy and apoptosis of autoreactive T cells | Production of autoantibodies<br>Proinflammatory cytokine production<br>Formation of ectopic germinal centers (EGCs) |

# Mechanisms linking fungal exposure and autoimmunity

Early inflammation helps contain invading pathogens, but uncontrolled or prolonged immune responses can hinder pathogen clearance and trigger off-target tissue damage, contributing to autoimmunity. A key question is whether and how immune interactions with fungi can precede or even drive autoimmunity. Here, we outline mechanisms by which fungus-host interactions may influence immune responses and susceptibility to autoimmunity through immune crosstalk with the mycobiome, molecular mimicry, epitope spreading, and immunomodulation.

## Mycobiome

Fungi coexist with bacteria and viruses in the mammalian microbiota, primarily in the gastrointestinal tract. Even though fungi comprise a minor part of the microbiota (Ott et al, 2008), they can exhibit significant diversity (Yan et al, 2024). Healthy human gut fungi predominantly belong to the phyla Ascomycota and Basidiomycota (Nash et al, 2017; Ott et al, 2008), including *Saccharomyces*, *Candida*, and *Malassezia* (Nash et al, 2017). Mice have also been observed to harbor commensal fungi such as *Candida tropicalis* and *S. cerevisiae* (Iliev et al, 2012). The gut mycobiota also has site-specificity, with mucosal fungi differing from luminal (fecal-associated) populations (Ott et al, 2008).

Mycobiota can compete with bacteria, with antibiotic depletion of bacterial commensals promoting fungal overgrowth and impacting immune responses. Although this review focuses on fungi, we recognize that fungal influences on host immunity often occur within the broader microbial ecological network. Notably, fungal–bacterial interactions, such as metabolic cross-feeding or competition in the gut, can shape the immune environment and inflammatory outcomes, adding an important layer of complexity to mycobiome–host interactions. For more detailed discussions on cross-kingdom interactions in health and disease, we refer readers to several excellent reviews on the topic (Iliev and Leonardi, 2017; MacAlpine et al, 2023; Miyauchi et al, 2023; Shirtliff et al, 2009).

### Gut fungi and host cell recognition

The gut mycobiota has been shown to play a critical role in mucosal immunity and eventually immunity in other organs (Fig. 1A). *C. albicans*, a key fungal resident of the gut, can drive protective and pathogenic immune responses depending on host–mycobiome interactions (Jiang et al, 2017; Shao et al, 2019). Intestinal epithelial cells and DCs can also recognize *C. albicans* and promote the differentiation and maintenance of Th17 cells, which are crucial for antifungal defense. Also, *C. albicans* hyphae secrete candidalysin, a cytolytic peptide toxin that damages epithelial cells and triggers IL-1β release, which amplifies IL-17 responses and neutrophil recruitment (Li et al, 2022b). CX3CR1[+] mononuclear phagocytes (MNPs), including DCs, macrophages, and monocytes, were identified as essential for developing Th17 responses to intestinal fungal colonization (Leonardi et al, 2018). These responses help contain fungal overgrowth but, when dysregulated, contribute to inflammation and barrier dysfunction.

### Mycobiota dysbiosis and fungal infections

Not only does mycobiota dysbiosis result from fungal infections, but it also contributes to further infections. Perturbations of

mycobiota—such as antibiotic use, immunosuppression, or epithelial barrier damage—can reduce host resistance and enable overgrowth or pathogenic transitions of fungi like *C. albicans* (Fan et al, 2015; Proctor et al, 2023). Expansion of such fungi can drive inflammation via secretion of virulence factors (e.g., candidalysin), activation of IL-17/IL-1β pathways, and disruption of epithelial integrity, creating a feed-forward loop of dysbiosis and inflammation, particularly in diseases like inflammatory bowel disease (IBD) (Li et al, 2022b; Moyes et al, 2016). Conversely, the invasive fungal infection itself can reshape the mycobiome by outcompeting other species or altering the immune landscape, leading to long-term ecological shifts and increased susceptibility to secondary infections or chronic inflammation.

### Fungal dysbiosis in the gut

Although IBD is not considered to be an autoimmune disease, its models supported a protective role of some gut fungi: antibiotic depletion of bacteria in mice led to immune dysregulation and susceptibility to severe colitis, but reintroducing *C. albicans* or *S. cerevisiae* restores barrier function and reduces inflammation (Jiang et al, 2017; Leonardi et al, 2022). Another study also demonstrated worsened colitis with prolonged antifungal treatment (Wheeler et al, 2016). In IBD patients, gut mycobiota generally shift towards reduced Ascomycota (including *Saccharomyces* and *Aspergillus* spp.) and increased Basidiomycota (such as *Malassezia* spp.) (Li et al, 2014; Liguori et al, 2016; Limon et al, 2019; Sokol et al, 2017) and *Candida* species (*C. albicans, C. tropicalis*) (Hoarau et al, 2016; Sokol et al, 2017). In addition, increased fungal diversity was linked to severe ulcerative colitis (UC) (Mar et al, 2016). Those shifts in mycobiota in patients raised intriguing possibilities for IBD treatments, but reverse causality cannot be ruled out. Further, longitudinal studies are necessary to determine temporal relationships. In addition, many studies have provided valuable insights into fungal community composition, but relatively few have incorporated functional analyses—such as metabolomic or transcriptomic profiling under reconstituted conditions with the fungal species of interest—which would be instrumental in clarifying the biological relevance of observed changes in the mycobiota.

### Fungal dysbiosis in host pathology of distant tissues

Mycobiome alterations appear to correlate with classical autoimmune disorders. A study indicated MS patients exhibiting increased gut *Saccharomyces* and *Aspergillus* (Shah et al, 2021), along with antifungal antibodies in the blood and cerebrospinal fluid (CSF) (Benito-León et al, 2010; Pisa et al, 2013). Increased *Candida*-specific enzyme activity was observed in MS patients (Saroukolaei et al, 2016), suggesting a potential fungal role in MS pathogenesis. Another study demonstrated the association of SLE to reduced fungal diversity, increased Ascomycota-to-Basidiomycota ratio, and *Candida* predominance (Li et al, 2022a; Yang et al, 2023). In SLE-prone mice, oral β-glucan administration accelerates disease (Fagone et al, 2014), possibly promoting strong Th17 responses. In addition, patients with RA demonstrated increased fecal *Candida* (Lee et al, 2022),

Besides those autoimmune diseases, fungal dysbiosis was also reported in various immune-mediated diseases. For example, in immune-mediated liver diseases (ILD), including primary biliary cholangitis, autoimmune hepatitis, and primary sclerosing cholangitis, elevated anti-*Saccharomyces cerevisiae* antibodies (ASCAs)

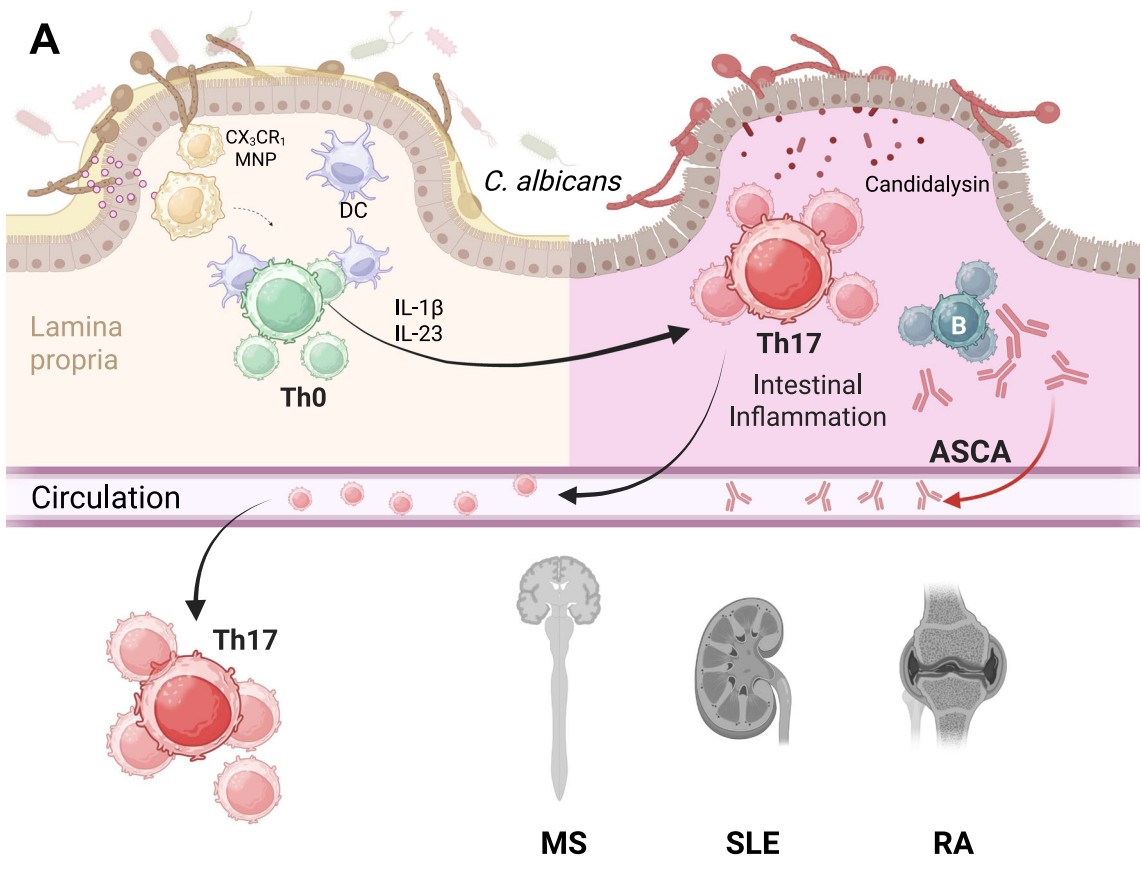

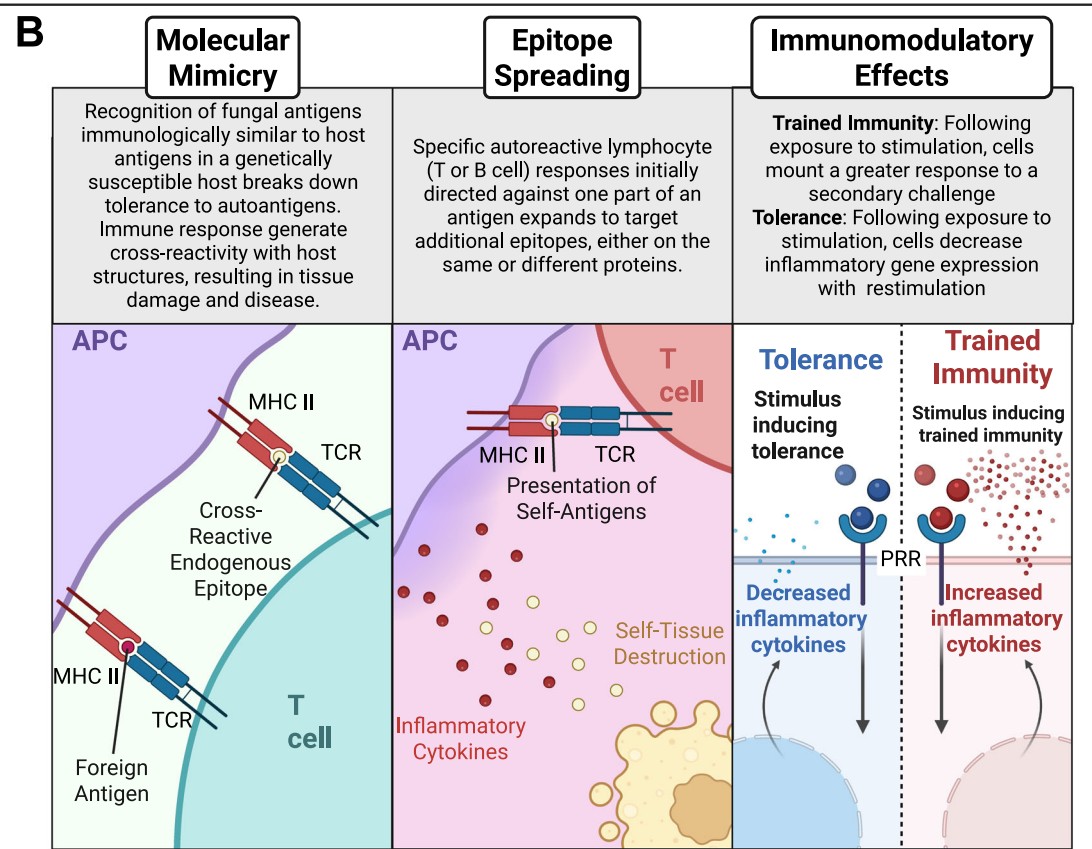

**Figure 1. Mechanisms that may link fungal infections to autoimmunity.**

(A) Fungal dysbiosis impacting adaptive immunity. Recognition of fungi by host cells influences epithelial integrity and immune homeostasis. Myeloid cells, primarily CX3CR1[+] MNPs, integrate fungal microbe-associated molecular patterns (MAMPs) and microenvironmental cues to coordinate immunity. Although commensal fungi can prevent inflammation, excessive Th17 responses contribute to autoimmunity. Candidalysin and proinflammatory cytokines, such as IL-1β, from myeloid cells are involved in Th17 polarization. Involvement of ASCAs is considered a possible candidate involved in pathogenicity. (B) Molecular and cellular mechanisms that may link fungal infections to autoimmunity. Although these links have yet to be fully established, the involvement of molecular mimicry, epitope spreading, and other immunomodulatory effects—such as trained immunity and innate tolerance—has been speculated. MNPs mononuclear phagocytes, ASCA anti-*S. cerevisiae* antibodies, DC dendritic cell, MS multiple sclerosis, RA rheumatoid arthritis, SLE systemic lupus erythematosus, APCs antigen-presenting cells, TCR T-cell receptor, MHC II major histocompatibility complex II. Graphics were created with BioRender.com.

suggested involvement of fungi (Muratori et al, 2003; Papp et al, 2010). In IBD-associated primary sclerosing cholangitis (PSC), the ratio of fungi to bacteria increased, and the presence of *Candida* in bile correlated with worse PSC outcomes (Lemoinne et al, 2020). Altered mycobiota and *Candida* β-glucan translocation into systemic circulation were suggested to exacerbate alcoholic cirrhosis (Yang et al, 2017). Notably, mice lacking *Clec7a* were protected from liver inflammation, suggesting that direct sensing of fungal β-glucans drives IL-1β-mediated hepatic inflammation (Yang et al, 2017). Patients with ankylosing spondylitis have also been linked to increased Ascomycota and decreased Basidiomycota (Berthelot et al, 2021). Patients with celiac disease were noted to display increased *Candida* and *Saccharomyces* colonization (Harnett et al, 2017).

### Fungal dysbiosis and fungal infection

Mycobiome dysbiosis can both promote and result from fungal infection. Under homeostatic conditions, commensal fungi such as *Candida* and *Malassezia* coexist with bacteria and host immune systems, contributing to immune tolerance and microbial balance. However, perturbations such as antibiotic use, immunosuppression, or epithelial barrier damage can reduce colonization resistance and enable overgrowth or pathogenic transitions of fungi like *C. albicans* (Fan et al, 2015; Proctor et al, 2023). The expansion of such fungi can drive inflammation through the secretion of virulence factors (e.g., candidalysin), activation of the IL-17/IL-1β pathways, and disruption of epithelial integrity, creating a feed-forward loop of dysbiosis and inflammation, particularly in diseases like IBD (Li et al, 2022b; Moyes et al, 2016). Conversely, invasive fungal infection itself can reshape the mycobiome by outcompeting other species or altering the immune landscape, leading to long-term ecological shifts and increased susceptibility to secondary infections or chronic inflammation.

### Shift of commensal to pathological fungi and its impacts on the host

Fungal species such *as C. albicans* and *Malassezia* spp. can shift from benign commensals to pathogenic organisms depending on host immune status and environmental factors. This balance can be disrupted by antibiotics, antifungal treatments, or changes in the local microenvironment. As a result, fungi that normally reside harmlessly on mucosal and skin surfaces may contribute to diseases when dysbiosis or immune dysfunction occurs—triggering inappropriate immune activation (Underhill and Iliev, 2014) and potentially promoting autoimmunity. Although yet to be clearly demonstrated in the context of autoimmune disease, evidence suggests that prolonged antifungal treatments can alter gut fungal populations and exacerbate conditions such as colitis and allergic airway disease (Wheeler et al, 2016). These effects were mediated in part by gut-resident CX3CR1[+] MNPs, which

facilitate crosstalk between intestinal fungi and host immunity (Leonardi et al, 2018; Li et al, 2018). These studies elegantly described a role for commensal fungi in shaping immune responses, but the causal mechanisms linking fungal dysbiosis to autoimmunity remain unclear. Future work should address whether the observed phenotypes are directly attributable to primarily to fungal pathogens or secondary immune alterations. In addition, although antifungal treatments are necessary, the broader effects of antifungals on host immunity are not yet fully understood and may introduce variables that complicate the interpretation of fungal causality.

To date, much of the research on fungal dysbiosis has focused on mucosal inflammatory diseases, including IBD and lung allergy. Yet, given the influence of mycobiota on immune responses, it is plausible that fungal dysbiosis also plays a role in autoimmunity—particularly through its potential impact on adaptive immunity. Thus, it would be important to clarify the possible link between mycobiota and autoimmunity when considering future treatment strategies for individuals suffering from autoimmune diseases.

## Molecular mimicry by fungi: suggested by homology but still lacking functional proof

Molecular mimicry occurs when microbial antigens resemble host molecules, potentially triggering autoreactive immune responses (Fig. 1B). In the context of fungal pathogens, evidence for molecular mimicry remains limited. So far, studies have primarily identified fungal epitopes through T-cell receptor (TCR) sequencing data from patients with autoimmune disorders (Grogan et al, 1999; Repac et al, 2021; Whalley et al, 2020). Fungal mimicry of self-antigens presents an intriguing potential mechanism in disease pathogenesis, but its contribution has yet to be firmly established. Most existing studies rely on sequence homology-based predictions with limited functional validation—such as demonstrating T- or B-cell cross-reactivity. Additional in vivo studies and immune profiling from patients will be critical to clarify the biological relevance of fungal mimicry in the development or progression of autoimmune diseases.

## Epitope spreading: a theoretical, unproven link between fungal infections and autoimmunity

Epitope spreading is an immune phenomenon where an immune response initially targets a specific epitope of an antigen and later broadens to recognize additional epitopes on the same or different proteins (Fig. 1B). It is particularly well-studied in animal models of autoimmune demyelination. In experimental autoimmune encephalomyelitis (EAE), an immune response initially targeting a myelin protein (e.g., myelin oligodendrocyte glycoprotein, MOG) later spreads to other myelin components (MBP, PLP) (Getts et al,

2013). In the Theiler's murine encephalomyelitis virus (TMEV) model, the initial virus responses to CNS infection led to host auto-reactivity to myelin epitopes, host tissue destruction, and myelin antigen presentation (Miller et al, 1997). To date, direct evidence for such classical epitope spreading in the context of fungal infections remains lacking. For example, even though the immunological aftermath of fungal infection may create a permissive environment for autoimmunity, this should be distinguished from true epitope spreading as mechanistically defined. Future studies using antigen-specific tools and longitudinal tracking of immune responses will be essential to determine whether fungal infections can drive bona fide epitope spreading, or whether their role is limited to bystander inflammation and antigen release.

## Immunomodulatory effects in innate immunity

In this subsection, we discuss how fungal recognition may shape local and systemic immunity through trained immunity (TI) and innate immune tolerance mechanisms of fungi (Fig. 1B). β-glucans can induce TI, a form of innate immune memory, where innate immune cells undergo long-term functional epigenetic reprogramming (Bek-kering et al, 2021; Netea et al, 2020). TI modulates cytokine expression by peripheral innate immune cells and reprograms bone marrow progenitor cells to enhance myelopoiesis and promote systemic inflammation. For example, a single intraperitoneal injection of β-glucan induces sustained myelopoiesis (Mitroulis et al, 2018) and aerobic glycolysis in myeloid cells (Cheng et al, 2014). These studies suggest that fungal infection could elevate systemic innate immune inflammation. Conversely, innate immune tolerance may also occur, driving myeloid cells toward an immunosuppressive state rather than heightened activation. In this process, repeated or prolonged exposure to certain fungal components could dampen inflammatory responses, reduce cytokine production, and limit immune activation (Lajqi et al, 2023). This shift in function can impair antifungal immunity and may contribute to increased susceptibility to secondary infections or tumor progression by suppressing protective immune responses.

## Outcomes in adaptive immunity

Numerous innate immune receptors sense fungi. Yet, Dectin-1 signaling in myeloid cells, shaping adaptive immunity, has been extensively studied in fungal infections. Ligating Dectin-1 with β-glucans promotes an antifungal immune response and protects mammalian hosts via triggering reactive oxygen species (ROS), phagocytosis, and proin-flammatory cytokine expression (Deerhake and Shinohara, 2021) at the levels of innate immunity, and ultimately promotes Th17 responses (LeibundGut-Landmann et al, 2007) (Fig. 1A). *C. albicans* is a well-characterized inducer of Th17 responses, which are critical for antifungal defense (Hernandez-Santos and Gaffen, 2012; LeibundGut-Landmann et al, 2007). However, Th17 responses can also contribute to immune-mediated pathology in genetically susceptible hosts or under dysregulated inflammatory conditions, as exemplified in autoimmune disorders such as MS and RA (Fig. 1A). This dual role of Th17 immunity highlights the context-dependent nature of fungal–host interactions and the importance of immune balance in determining disease outcomes.

Although fungal infections typically elicit proinflammatory immune responses, β-glucan exposure can also regulate immune responses. For example, Dectin-1 signaling mitigates EAE devel-opment via a neuroprotective mechanism through a CARD9-independent pathway (Deerhake et al, 2021a). Intravenous injec-tion of zymosan-depleted, a Dectin-1-specific agonist, delays the disease course (Deerhake et al, 2021a). β-glucan also delays T1D onset in murine models (Karumuthil-Melethil et al, 2014; Karumuthil-Melethil et al, 2008; Taylor and Vasu, 2021) by promoting the production of anti-inflammatory cytokine expres-sion and increasing Treg frequency. Research is ongoing to understand how the β-glucans/Dectin-1 axis can be either proinflammatory or regulatory. At least, a current explanation is that functional outcomes depend on distinct signaling pathways downstream of Dectin-1 (Deerhake et al, 2021a; Deerhake and Shinohara, 2021).

Our discussion also encompasses antifungal antibodies: ASCAs serve as serological markers for CD (Main et al, 1988) and are found in patients with autoimmune disorders, such as SLE (Dai et al, 2009a; Mankai et al, 2013), T1D (Sakly et al, 2010), and RA (Dai et al, 2009b). Although "ASCAs" were named after *S. cerevisiae*, the fungus was rarely isolated from stools and mouth swabs from patients, including those with CD (Sendid et al, 2024). Instead, primary fungal sources of ASCA epitopes are now considered to be *Candida* spp. (Muller et al, 2010; Schaffer et al, 2007; Sendid et al, 2024). Although the precise mechanism remains under investigation, ASCAs may contribute to inflammation, particularly in the gut, through multiple interacting factors.

These studies highlight how contextual cues in adaptive immunity shape immune responses to fungi in autoimmune diseases. Yet, a more comprehensive understanding of immunomodulatory fungal factors may be gained by incorporating physiologically relevant conditions, considering host-related variables such as genetic susceptibility and microbiota composition, and further clarifying how cytokine responses, such as IL-17 induction, contribute to either protective immunity or pathological outcomes.

# Studies highlighting autoimmune diseases and fungal infections

Evidence increasingly connects autoimmune diseases to fungal infec-tions, which were implicated in developing autoimmune diseases and complications from immunomodulatory treatments (Fig. 2).

## MS

Studies revealed a possible association between invasive fungal infections (IFIs) and MS. Fungal antigens and antifungal antibodies have been detected in the blood and CSF of MS patients (Alonso et al, 2018; Benito-León et al, 2010; Pisa et al, 2013). A report indicated that MS patients showed a significantly higher seroprevalence of antibodies against *Candida* species compared with healthy controls (Benito-León et al, 2010). The HLA-DRB1*15:01 allele, a major genetic risk factor of MS (Schmidt et al, 2007), is also associated with susceptibility to allergic bronchopulmonary aspergillosis (ABPA) (Chauhan et al, 2000; Nasir et al, 2023). MS patients, particularly those with relapsing-remitting MS (RRMS), exhibit alterations in their gut mycobiome (Yadav et al, 2022), including an increased fungal-to-bacterial ratio (Yadav et al, 2022) and a greater abundance of *Aspergillus* and *Saccharomyces* (Shah et al, 2021).

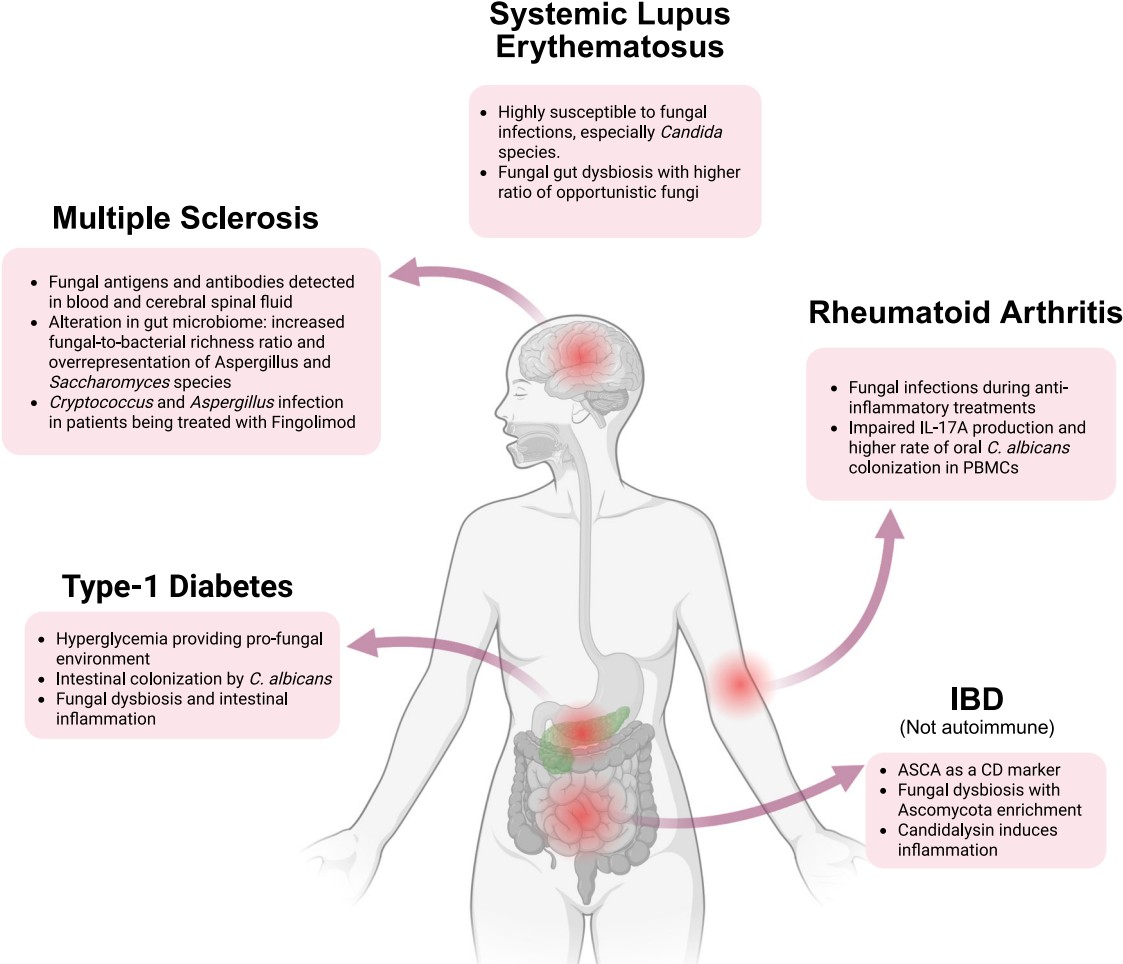

**Systemic Lupus Erythematosus**

- Highly susceptible to fungal infections, especially *Candida* species.
- Fungal gut dysbiosis with higher ratio of opportunistic fungi

**Multiple Sclerosis**

- Fungal antigens and antibodies detected in blood and cerebral spinal fluid
- Alteration in gut microbiome: increased fungal-to-bacterial richness ratio and overrepresentation of Aspergillus and *Saccharomyces* species
- *Cryptococcus* and *Aspergillus* infection in patients being treated with Fingolimod

**Rheumatoid Arthritis**

- Fungal infections during anti-inflammatory treatments
- Impaired IL-17A production and higher rate of oral *C. albicans* colonization in PBMCs

**Type-1 Diabetes**

- Hyperglycemia providing pro-fungal environment
- Intestinal colonization by *C. albicans*
- Fungal dysbiosis and intestinal inflammation

**IBD**
(Not autoimmune)

- ASCA as a CD marker
- Fungal dysbiosis with Ascomycota enrichment
- Candidalysin induces inflammation

**Figure 2.  Autoimmune conditions associated with fungal infections.**

Various autoimmune and inflammatory conditions are impacted by fungal infections through fungal gut dysbiosis, antifungal antibodies and antigens, and immunomodulatory treatment. We summarized cases on RA, SLE, MS, T1D, and IBD (which is not an autoimmune disease). Graphics were created with BioRender.com.

The majority of current MS treatments target the immune system to reduce central nervous system (CNS) inflammation. For example, fingolimod, a sphingosine-1-phosphate receptor agonist, restricts T-cell infiltration into the CNS, but T cells are crucial for the host defense against *Cryptococcus* infection in the brain (Uicker et al, 2006). Indeed, fingolimod was linked to increased susceptibility to IFIs by *Cryptococcus* and *Aspergillus* (Scotto et al, 2021). Thus, it is not surprising that the use of fingolimod was further associated with an increased risk of cryptococcal meningoencephalitis (CM) (Cuascut et al, 2021; Nasir et al, 2023).

### RA

RA patients face an elevated risk of IFIs. However, fewer articles on this aspect are available compared with MS. Interestingly, peripheral blood mononuclear cells (PBMCs) from RA patients showed impaired *Candida*-induced IL-17A production, along with a higher rate of oral *C. albicans* colonization, despite increased baseline IL-17A levels (Bishu et al, 2014). This paradox suggested that dysregulated Th17 responses may impair antifungal immunity,

potentially due to an imbalance between protective and pathogenic Th17 cell subsets. Reports also indicated that anti-inflammatory treatments for RA can increase the risk of fungal infections (Chen et al, 2017; Pieta et al, 2023).

### T1D

The link between T1D and IFIs, particularly *C. albicans*, has been primarily reported in the context of hyperglycemia. Previous studies showed that elevated blood glucose levels provided a favorable environment for fungal growth, as they also impaired neutrophil-mediated phagocytosis and weakened antifungal immunity (Rodrigues et al, 2019; Wilson and Reeves, 1986).

### SLE

SLE patients are at high risk for IFIs due to immune dysfunction and immunosuppressive therapies. Risk factors for IFIs included severe disease phenotype, glucocorticoid use, and immunosuppressants (Meng et al, 2023). Mortality in SLE patients with IFIs was

linked to leukopenia and high-dose glucocorticoids (Martinez-Martinez et al, 2012). *Candida* species were the most frequently reported fungal pathogens, particularly in corticosteroid-treated and juvenile patients (Su et al, 2021; Tanveer et al, 2024). Diagnosing IFI in SLE is challenging as symptoms mimic lupus flares, delaying treatment (He et al, 2021).

### IBD

Although IBD is not considered a classical autoimmune disease, we discuss IBD due to its aberrant immune responses, where innate and adaptive immune responses contribute to inflammation. *Candida* species are significantly more abundant in IBD patients across various geographic regions (Chehoud et al, 2015; Imai et al, 2019; Liguori et al, 2016; Sokol et al, 2017). Similarly, fungal dysbiosis exacerbated colonic mucosal inflammation in IBD, with an enrichment of Ascomycota as observed in CD patients (Liguori et al, 2016). Thus, fungal dysbiosis was suggested as a key component in IBD pathogenesis (Imai et al, 2019). Recent research highlighted the role of fungal strain-specific immune responses, such as candidalysin from *C. albicans*, in driving IL-1β-mediated inflammation, further suggesting fungi as pivotal contributors to IBD pathology (Li et al, 2022b). However, *NOD2* gene mutations, commonly linked to CD, do not appear to significantly impact mycobiota composition (Nelson et al, 2021). Here, critical knowledge gaps remain, including the precise molecular mechanisms by which specific fungal strains (e.g., *C. albicans* and *C. tropicalis*) contribute to IBD pathology and whether targeted antifungal therapies improve disease outcomes without disrupting immune homeostasis.

### Risk and challenges in autoimmunity and IFIs

The elevated risk of IFIs in patients receiving immunomodulatory therapies—combined with the challenges of detection and treatment—highlighted the need for improved strategies to identify and monitor these infections. Although recent advances have explored blood biomarkers for IFI detection (including detection of fungi by PCR or identification of antigens), the gold-standard continues to rely on clinical suspicion coupled with evidence from microscopy or serial blood cultures (Cornely et al, 2025; Galmiche et al, 2023; Taynton et al, 2022). Though essential, these traditional methods are often time-consuming, require specialized expertise, and may delay appropriate treatment. Broader adoption, refinement, and validation of novel diagnostic tools will be critical to enable more timely and accurate detection of fungal infections, particularly in vulnerable patient populations.

At the same time, there is increasing interest in the potential connection between fungal infections and the development of autoimmune diseases. Although this area of research is still in its early stages, emerging studies have suggested temporal associations between fungal infections and autoimmune onset. However, experimental models used to study fungal infection often rely on high inoculum doses that do not reflect natural exposure and may not precisely replicate the kinetics or localization of human disease—thereby limiting translational relevance. Furthermore, establishing a direct causal link remains challenging, as confounding factors such as pre-existing immune dysregulation or genetic susceptibility complicate interpretation. Addressing these limitations through improved modeling and mechanistic investigation will be essential to better understand the complex interplay between fungal pathogenesis and autoimmunity.

Notably, despite the advances in understanding host–fungal interactions, clinical recognition of IFIs in patients with immune-mediated diseases remains challenging. In addition to conventional diagnostics, molecular and biomarker-based assays (e.g., galactomannan, β-glucans, and fungal genetic taxonomy) are under development and show promise for earlier detection (Fang et al, 2023). Validation of these approaches in immunomodulated autoimmune cohorts and developing personalized thresholds to guide actionable surveillance will be important next steps.

## Implications of antifungal therapeutics on autoimmunity

Antifungal treatments primarily target fungi, but they could affect host immunity. As a result, alteration in immune-fungal interactions and triggering off-target immune activation could happen. Below, we examine antifungal therapies and their implications for immune regulation.

### Current antifungal therapies

FDA-approved antifungal therapies primarily fall into four classes: (1) polyenes (e.g., Amphotericin B), (2) Azoles (e.g., Fluconazole, voriconazole), (3) pyrimidine analogs (e.g., flucytosine (5-FC), and (4) echinocandins (e.g., Caspofungin) (Fig. 3A). Given the eukaryotic homology between fungi and humans, it is crucial to understand if antifungals trigger cross-reactivity to host cells and modulate the immune system.

Polyenes, such as Amphotericin B, disrupt fungal membranes but can also affect mammalian cells by targeting cholesterol (Readio and Bittman, 1982) and possible DAMP release. As microbe-derived molecules, polyenes engage CD14, TLR2, and TLR4 on phagocytes and trigger immune responses too (Bellocchio et al, 2005; Sau et al, 2003). Azoles block ergosterol biosynthesis, compromising fungal membrane integrity, but also enhance inflammatory gene expression, especially *Tnf*, in immune cells in the absence of fungi (Fidan et al, 2014; Simitsopoulou et al, 2008). Pyrimidine analogs, including 5-FC, are used primarily with Amphotericin B or azoles. After entering fungal cells, 5-FC is converted to 5-fluorouracil (5-FU), which inhibits fungal DNA and RNA synthesis. Direct evidence linking 5-FU to autoimmunity is still unknown, but its immunomodulatory effects, such as cytotoxicity on myeloid-derived suppressor cells (MDSCs), were reported in a cancer setting (Vincent et al, 2010). Echinocandins inhibit fungal cell wall synthesis and unmask β-glucans for immune recognition to promote immune responses (Lamaris et al, 2008; Salvenmoser et al, 2010; Wheeler et al, 2008). Antifungal therapies, though effective against IFIs, can also provoke immune activation and DAMP release, which may inadvertently trigger bystander immune responses and self-antigen recognition, contributing to the onset or aggravation of autoimmunity.

### Potential immunomodulatory strategies for antifungal treatments

With the rise of drug-resistant fungi, alternative therapies focus on enhancing immune responses (Fig. 3B) rather than directly targeting fungi. However, their potential to induce autoimmunity remains to be explored.

## A

### Antifungal Therapy Mechanisms

### Off-Target Effects

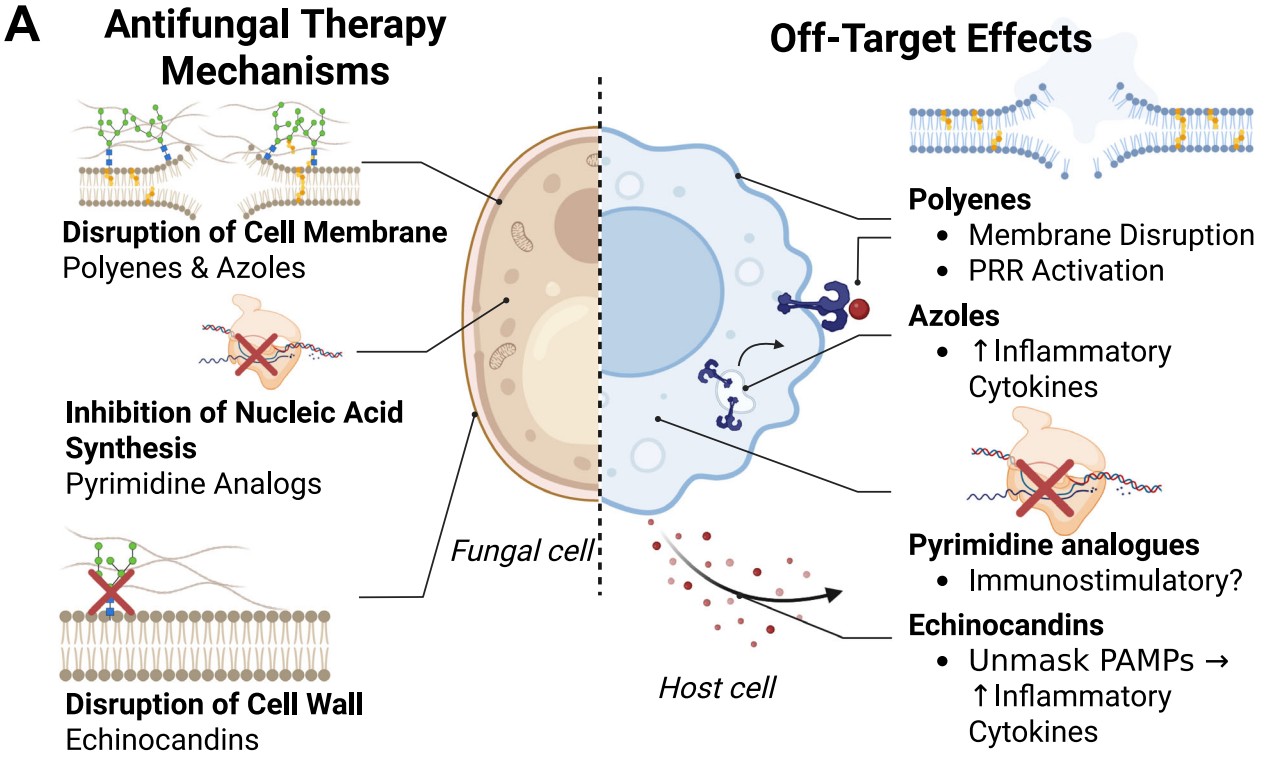

**Disruption of Cell Membrane**
Polyenes & Azoles

**Inhibition of Nucleic Acid Synthesis**
Pyrimidine Analogs

**Disruption of Cell Wall**
Echinocandins

*Fungal cell*

*Host cell*

**Polyenes**
- Membrane Disruption
- PRR Activation

**Azoles**
- ↑Inflammatory Cytokines

**Pyrimidine analogues**
- Immunostimulatory?

**Echinocandins**
- Unmask PAMPs → ↑Inflammatory Cytokines

## B

### Biologic Therapies

**Colony Stimulating Factors/Cytokines**
- G-CSF
- M-CSF
- GM-CSF
- IFNγ

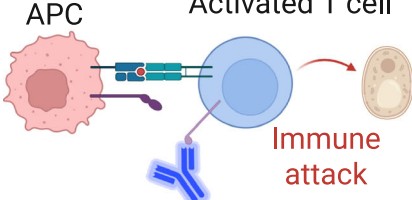

**Checkpoint Inhibition**

APC          Activated T cell

Immune attack

Checkpoint Inhibitor

**Monoclonal Antibodies**

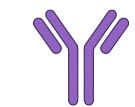

**Anti-Fungal Vaccination**

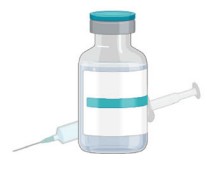

### Cell-Based Therapies

**Granulocyte Transfusion**

Granulocytes From ABO-Compatible Donors

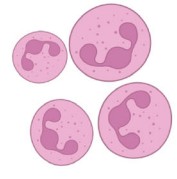

**Anti-Fungal CAR-T Immunotherapy**

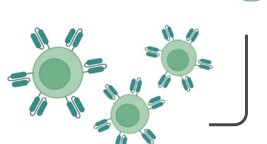

CAR T cells

◀ **Figure 3.  Antifungal therapeutics, immune modulation, and autoimmunity.**

(A) Antifungal therapies may impact autoimmune processes by modulating immune responses. Standard antifungal drugs—including polyenes, azoles, pyrimidine analogs, and echinocandins—not only target fungal components but can also activate innate immune receptors or allow host DAMP release, potentially triggering off-target immune activation. (B) As new strategies emerge, immunomodulatory treatments like colony-stimulating factors, IFNγ, and immune checkpoint inhibitors show promise in enhancing fungal clearance but may carry risks of promoting or exacerbating autoimmunity. Likewise, biologics and vaccines are advancing but require further investigation to assess long-term immune effects. Cell-based therapies, including granulocyte transfusions and fungal antigen-specific CAR-T cells, offer novel antifungal approaches but raise concerns about T-cell dysregulation. Graphics were created with BioRender.com.

### Biologic therapeutics

Colony-stimulating factors (CSFs), such as G-CSF, M-CSF, and GM-CSF, enhance myelopoiesis and fungal clearance (Bandera et al, 2008; Du et al, 2020; Nemunaitis et al, 1993), but GM-CSF is particularly linked to autoimmune diseases as a detrimental factor in MS (Codarri et al, 2011; El-Behi et al, 2011; McQualter et al, 2001). IFNγ is also well-studied in its antifungal effect. IFNγ can limit Th17 polarization (Harrington et al, 2005; Zhang, 2007) and could support Tregs (Wang et al, 2006), but it may also contribute to autoimmunity. Immune checkpoint inhibitors (ICIs) are used in cancer treatments, but they also have shown promise in enhancing fungal clearance preclinically (Chang et al, 2013; Vu et al, 2020; Wurster et al, 2020). At the same time, how immune-related adverse events with ICIs could impact hosts during fungal infections will need further research. Antifungal monoclonal antibodies and vaccines have been explored as strategies to treat and prevent IFIs, showing promising results in preclinical studies (Bromuro et al, 2010; Bugli et al, 2013; Pachl et al, 2006). However, limitations in our understanding of their functional and mechanistic links to autoimmunity remain a key barrier to clinical translation. In addition, as biologic therapeutics emerge as promising antifungal treatments, it is also crucial to develop regimens that optimize antifungal efficacy but minimize the risk of excessive inflammation and autoimmunity.

### Cell-based therapies

Adoptive cell transfer has been investigated to treat IFIs in immunocompromised patients. Granulocyte transfusions, often enhanced with G-CSF (Grigull et al, 2006; Grigull et al, 2002), have shown limited efficacy (West et al, 2017). As fungal antigen-specific T cells may offer protection against fungal infections (Gottlieb et al, 2021; Papadopoulou et al, 2016; Tramsen et al, 2013), a recent preclinical study also demonstrated the efficacy of antifungal chimeric antigen receptor (CAR) T cells to treat chronic pulmonary aspergillosis (Seif et al, 2022). Both granulocytes (as phagocytes and major producers of proinflammatory cytokines, proteases, and ROS) and CAR-T cells (through cytokine release, fungicidal activity, and the production of proinflammatory cytokines) have the potential to exacerbate systemic inflammation that may activate autoreactive T and B cells. Although the development of fungal antigen-targeted cell-based therapies is exciting, further studies are necessary to examine the risk of triggering or exacerbating autoimmunity by potential off-target effects, tissue damage, T-cell dysregulation, and possible epitope spreading.

## Conclusion

The relationship between fungal infections and autoimmunity is complex and bidirectional: Fungal pathogens can disrupt immune homeostasis and contribute to autoimmune processes; conversely, autoimmunity and its treatments increase susceptibility to fungal infections. This dynamic interplay presents challenges for clinical management and underscores the need for integrated therapeutic strategies. The emergence of drug-resistant fungi further highlights the urgency of advancing our understanding of antifungal immunity and its implications for autoimmune risk. Future studies should address how antifungal therapies influence immune responses and clarify how pre-existing autoimmune conditions modify antifungal immunity. By elucidating fungal virulence mechanisms, host immune pathways, and the immunological consequences of antifungal interventions, we can develop targeted approaches to improve outcomes and gain broader insights into immune regulation in health and disease.

## Pending issues

- What are the mechanistic links between specific fungal antigens and autoimmune responses?
- How do antifungal therapies alter immune homeostasis in patients with or at risk for autoimmunity?
- What are the long-term immunological consequences of mycobiome shifts in autoimmune and immunocompromised individuals?
- How do host genetics and microbiome context modulate susceptibility to fungal-triggered autoimmunity?
- Can fungal strain-specific immune signatures be leveraged for predictive diagnostics or targeted interventions?
- What are the risks and benefits of immunomodulatory antifungal strategies (e.g., biologics, CAR-T, checkpoint inhibitors) in individuals predisposed to autoimmunity?
- How can experimental models be refined to better reflect physiological fungal exposure and chronic infection relevant to human autoimmunity?

## Peer review information

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

## Acknowledgements

MLS received funds from NIH (R01-AI088100, R01-NS120417, R01-AI160737); MKL is supported by the National Science Foundation (NSF) (DGE-2139754).

## Author contributions

**Devon T DiPalma**: Data curation; Validation; Investigation; Writing—original draft. **Miranda K Lumbreras**: Data curation; Validation; Investigation; Writing—original draft. **Mari L Shinohara**: Conceptualization; Supervision; Validation; Investigation; Visualization; Project administration; Writing—review and editing.

## Disclosure and competing interests statement

MLS currently receives funds from Ono Pharmaceuticals, Co. Ltd. to unrelated topics covered in this review.

