## [Peer Review File · EMBO Molecular Medicine]

Interplay Between Fungal Infections and Autoimmunity: Mechanisms and Therapeutic Perspectives

Devon DiPalma, Miranda Lumbreras, and Mari Shinohara

Corresponding author: Mari Shinohara (mari.shinohara@duke.edu)

Review Timeline:

Submission Date:	20th Mar 25
Editorial Decision:	28th Mar 25
Revision Received:	28th May 25
Accepted:	2nd Jun 25

Editor: Zeljko Durdevic

Transaction Report:

28th Mar 2025

Dear Dr. Shinohara,

Thank you for the submission of your manuscript to EMBO Molecular Medicine. We have now received feedback from the two reviewers who agreed to evaluate your manuscript. As you will see from the reports below, the referees are positive about its interest and timeliness, however, they also raise serious criticisms that should be fully addressed in a revised manuscript.

I would also like to ask you to amend the following:

- 1) Add up to 5 keywords.
- 2) Figure 1 and Figure 4B would be better suited as Tables. If BioRender was used to create the figures, please add following sentence to the figure legends: "Graphics were created with BioRender.com."
- 3) Please provide more detailed description of all figures in the legends.
- 4) Please add "Glossary". The glossary is meant to explain some of the terms used for laymen. Could you please identify terms that may need an "explanation"?
- 5) Please add "Pending issue". At the end of each article is a box highlighting issues that still need further studies and where research efforts should converge. Could you identify some pending issues?
- 6) Rename "Competing interests" to "Disclosure and competing interests statement". We updated our journal's competing interests policy in January 2022 and request authors to consider both actual and perceived competing interests. Please review the policy <https://www.embopress.org/competing-interests> and update your competing interests if necessary.
- 7) Place "Disclosure and competing interests statement" and "Acknowledgments" before references.
- 8) Correct the reference citation in the text and reference list. In the text, a reference should be cited by author and year of publication. Include a space between a word and the opening parenthesis of the reference that follows. In the reference list, citations should be listed in alphabetical order. Where there are more than 10 authors on a paper, 10 will be listed, followed by "et al.". Please check "Author Guidelines" for more information.
<https://www.embopress.org/page/journal/17574684/authorguide#referencesformat>
- 9) Author contributions: Please remove it from the manuscript and specify author contributions in our submission system. CRediT has replaced the traditional author contributions section because it offers a systematic machine-readable author contributions format that allows for more effective research assessment. You are encouraged to use the free text boxes beneath each contributing author's name to add specific details on the author's contribution. More information is available in our guide to authors:
<https://www.embopress.org/page/journal/17574684/authorguide#authorshipguidelines>
- 10) As part of the EMBO Publications transparent editorial process initiative EMBO Molecular Medicine will publish online a Review Process File (RPF) to accompany accepted manuscripts. This file will be published in conjunction with your paper and will include the anonymous referee reports, your point-by-point response and all pertinent correspondence relating to the manuscript. Let us know whether you agree with the publication of the RPF.

I hope that the referees' comments do not prove too problematic to address and I look forward to reading your next version.

Yours sincerely,

Zeljko Durdevic

*** IMPORTANT INFORMATION ***

- 1) a .doc formatted version of the manuscript text (including Figure legends and tables)
- 2) Separate figure files
- 3) a letter INCLUDING the reviewer's reports and your detailed responses to their comments.

Also, and to save some time should your paper be accepted, please read below for additional information regarding some features of our research articles:

1) Glossary: EMBO Molecular Medicine articles will be accompanied by a glossary explaining some of the terms used for laymen. I identified the following:

_____, _____, _____

Could you please help us in identifying terms that may need an "explanation" other terms that we can add to the glossary.

2) Pending issues: At the end of each article we will have a box highlighting issues that still need further studies and where research efforts should converge (we call this the Pending issues box). From my reading I would say:

but I can see there may be many more. Could you work on this as well?

3) Disclosure and competing interest statement: Please include a statement declaring any competing commercial interests in relation to your submitted work.

4) Please note that we now mandate that all corresponding authors list an ORCID digital identifier. This takes <90 seconds to complete. We encourage all authors to supply an ORCID identifier, which will be linked to their name for unambiguous name identification.

Currently, our records indicate that the ORCID for your account is 0000-0002-6808-9844.

Link Not Available

-

Thank you,

Zeljko Durdevic

***** Reviewer's comments *****

Referee #1 (Remarks for Author):

This review article provides a broad overview of the bidirectional relationship between fungal infections and autoimmunity, highlighting the key mechanisms and therapeutic implications. This is a timely topic that will be of use and interest to a broad scientific audience. The organization by autoimmune and inflammatory conditions is particularly informative and different from other reviews on this topic. The figures are also detailed, though several of the words and icons are very small and difficult to see. My main critiques are outlined below.

(1) There's one issue not fully discussed in this review is the relationship between fungal infection and commensal fungi in the mycobiome. It's always good to discuss about the immune crosstalk with mycobiome in autoimmunity. But it's unclear whether the commensal-pathogen transition contributes to autoimmunity. Furthermore, how does the mycobiome dysbiosis contributes to fungal infection or vice versa?

(2) In several studies discussed in this review, the claims do not align with the presented data. The authors should discuss such

shortcomings rather than solely highlighting the studies own claims. Shortcomings should be discussed, especially in such a nascent area of research.

(3) While the review extensively discusses fungal pathogens and the mycobiome, recent advances emphasize that the effects of fungi on host immunity are modulated through interactions with other microbes, including bacteria, viruses, and archaea, within the microbial ecological network. For example, synergistic or antagonistic cross-kingdom interactions (e.g., fungal-bacterial metabolic crosstalk in the gut) can significantly influence immune responses. Also, fungal dysbiosis may exacerbate or mitigate autoimmune processes, largely related to the composition of the bacterial community. Incorporating these aspects would provide new insights into the fungal-autoimmune interactions within the broader ecological network of the integral microbiota.

(4) In addition, the review predominantly frames fungi as drivers of autoimmunity but does not sufficiently address their potential protective or regulatory roles. To provide a more balanced perspective, it is suggested to highlight the dual effects of commensal fungi. For instance, *Candida albicans* promotes Th17 responses that protect against pathogens but may also exacerbate inflammation in susceptible hosts.

(5) Finally, clinical monitoring strategies for immunosuppressed patients are also important. The review acknowledges the increased risk of fungal infections in autoimmune patients undergoing immunomodulatory therapies but lacks discussion on actionable clinical guidance on how to manage this risk. For example, biomarker-based surveillance as well as personalized thresholds.

(6) Figures: Provide a list of fungal species, either commensal or pathogenic, in different autoimmune diseases. Moreover, I would suggest a figure describing the triple interaction of "fungal infection - mycobiome - autoimmunity".

Addressing these points would significantly enhance the manuscript by incorporating fungal-autoimmune interactions within a multispecies ecological model, providing a more balanced view of fungal roles, clarifying the therapeutic risks associated with emerging biologic therapies, and offering practical, personalized clinical recommendations. The revisions would better align the review with current trends in microbiome research and precision medicine, ultimately increasing its impact and clinical relevance.

Referee #2 (Remarks for Author):

In this review article, the authors described the physiological (beneficial) and pathological (detrimental) roles of fungi and antifungal immune responses. The review summarized the relationships between representative fungi and major immune cells, discussing the potential risk and mechanism(s) of induction of immune-mediated diseases by fungal infections; these topics have rarely been reviewed by others. Thus, this reviewer recommends the manuscript for publication. In this manuscript, however, several issues need to be addressed: for example, the inaccurate definition of several terms and the overemphasis of several findings that were only loosely associated with fungal infections.

Major concerns

1. The authors used the present tense in most sentences in this manuscript. This is incorrect; most experimental and clinical findings should be described using past or present perfect tense. For example, in section 4, most sentences were based on experimental and clinical findings, thus should be described in the past tense (if the finding was based on a few manuscripts) or the present perfect tense (if the finding was based on several manuscripts in the past 10 years or so). Present tense usage gives the reader the wrong impression that the description was established as a permanent truth. Other examples of misusages of the present tense are as follows: on line 49, "linked" instead of "links". All sentences on lines 202-212.

2. The usage of several terms, including "autoimmune disease" and "autoinflammatory disease," was inaccurate (e.g., line 203). As the authors mentioned, inflammatory bowel disease (IBD) is not categorized as an autoimmune disease. There are other diseases in this manuscript, which are not categorized as autoimmune diseases: spondyloarthritis, celiac disease (line 231), and animal models on lines 220-222. In general, immune cells in acquired immunity play a major role in autoimmune diseases; activation of innate immune cells causes (hereditary) autoinflammatory diseases (e.g., on line 265, "autoinflammatory disease development" should be used, instead of "autoimmune disease development." Here, it should be noted that an elevated innate immunity can enhance, but not prime, autoimmune responses.). The authors misused "autoinflammatory" as "autoimmune" in the text. This reviewer recommends using the term "immune-mediated" instead of "autoimmune" in most sentences in this article, including the title (e.g., lines 143, 164). The term "immune-mediated diseases" includes autoimmune diseases, autoinflammatory diseases, and other immune-mediated diseases with unknown etiology or induced by microbial infections. Thus, the term "immune-mediated disease" is more appropriate than "autoimmune diseases" in this article.

3. In this manuscript, the two substantial pieces of evidence linking fungal infections and autoimmune diseases are found in ASCA and beta-glucan, although associations of most other findings with autoimmune diseases cited in this manuscript were not strong, with little supportive evidence. The findings of ASCA and beta-glucan were described in several sections, often redundantly (e.g., lines 95, 156, 216, 334). Thus, this reviewer recommends setting up two new paragraphs, each describing the findings of ASCA and beta-glucan with more explanations as the most strong evidence linking fungal infections and autoimmunity.

4. Section 3b) Molecular mimicry. In bacteria and virus infections, antibodies, not T cells, have been reported to recognize cross-reactive epitopes by molecular mimicry, for example, anti-glycolipid antibodies and *Campylobacter jejuni* in Guillan Barre syndrome or brain antigens and EB virus in multiple sclerosis. On the other hand, in fungal infections, T cells seemed to recognize cross-reactive epitopes, according to this manuscript. Are there any examples of cross-reactive antibodies in fungal infections? If not, it is good to speculate/discuss why cross-reactive T cells, but not antibodies, have been reported in fungal infections, unlike bacteria or viruses (e.g., differences in complement activation, role of T cells, plasma membrane structure, etc).

5. Section 3c) epitope spreading. Although the authors cited EAE as an example of epitope spreading of EAE, a more appropriate citation should be epitope spreading in microbial infections. For instance, in reference 96, Stephen Miller's group described epitope spreading in a viral model of multiple sclerosis, Theiler's murine encephalomyelitis virus (TMEV) infection in mice. In the TMEV model, T cells derived from TMEV-infected mice recognized TMEV antigens, initially. However, inflammation in the spinal cord resulted in the release of myelin antigen, leading to epitope spreading to PLP139-151, which TCR recognizes. Later, epitopes further spread to PLP171-191, followed by MBP84-104, and lastly to MOG96-102. In TMEV infection, although direct virus infection also causes myelin destruction, it is not regarded as the primary cause of epitope spreading. In microbial infections, the definition of epitope spreading is the changes of the immune epitope from microbial epitope to host epitope; this has been described in other diseases, including rheumatic disease and experimental HSV infection. Thus, in fungal infection, the term "epitope spreading" should be used only when fungal epitopes recognized by TCR or BCR spread to host epitopes recognized by TCR or BCR. Induction of autoimmune cells by the release of host proteins following direct fungal infection in the organ should not be called epitope spreading. Thus, the entire paragraph of section 3c) should be rewritten.

Minor concerns

1. The authors sometimes did not cite the key references that should be required for understanding the manuscript's contents. In Section 1, there were no single references. At least, a sentence on lines 50-51 needs references. The following were lines that need references: lines 75, 158.

2. Line 85, "people" instead of "those"

3. Line 88, "MS patients" instead of "patients"

4. Line 89, "will be discussed" instead of "are discussed"

5. Line 93, "Crohn's disease" instead of "Crohn's Disease"

6. Line 93, "SLE" should be spelled out here, instead of lines 226-227. The following terms should be also spelled out when the terms were used for the first time: IL-23 (line 108), TNF (line 108), Th (line 109), ROS (line 111), IFN (line 128), UC (line 201)

7. Line 102, delete "but not limited to"

8. The ambiguous word "while" is overused; in most sentences, it can be replaced with the more specific word "although," for example, on lines 122, 155, and 172

9. Line 126, "(NK) cells" instead of "(NK)

10. Line 146, "Th" instead of "T helper"

11. Line 153, "cellular cytotoxicity" instead of "cell toxicity"

12. Line 173, "human gut" instead of "gut"

13. Line 178, "correlative" is unclear. This sentence needs to be paraphrased.

14. Line 184, "MNPs" need more explanation, such as their phenotype and origin.

15. Line 196, "inflammatory bowel disease" instead of "Inflammatory Bowel Disease"

16. Line 203, "IBD" instead of "autoinflammatory disease"

17. Line 212, "IBD" instead of "inflammatory disease"

18. Line 216, there are several "autoimmune liver disease." This needs more specification.

19. Line 230: Since there are several types of spondyloarthropathy, a more specific description or explanation is required here.
20. Line 307, "compared with" instead of "compared to"
21. Line 365, "enhance inflammatory genes in immune cells" needs more explanations.
22. Line 368, "its immunostimulatory effects" needs more explanations.
23. Lines 371-373. This sentence does not make sense; the sentence should be rewritten or deleted.
24. Line 387, "immunotoxicity risks" need more explanations.
25. Line 388 and figure 4, "Monoclonal antibodies" need explanations. Are they antifungal antibodies or some other antibodies?
26. Line 388, "knowledge gaps". Are there any reports of induction of autoimmunity in the treatment described here?
27. Lines 400-401. This sentence needs to be rewritten. The authors need to discuss more specifically how G-CSF or CAR-T cell treatment potentially causes autoimmune diseases.
28. The conclusion section is too long. It should be written in one paragraph with a more explicit message.
29. Figure 1. The following 1) to 4) were not explained in the text. Thus, the authors should delete the phrases or include the explanations in the text. 1) Neutrophil: contribution of chronic inflammation; promotes autoimmunity in leukocytes. 2) Dendritic cell: pathogen containment by granuloma formation; May promote immune exhaustion. 3) T cell: granuloma formation. 4) B cell: formation of ectopic germinal centers. In addition, ROS, NETs, PRRs, Th should be spelled out in figure legend.
30. Figure 2. Myeloid NMP was not explained well in the text. Thus, more explanations on NMP should be included in the text, or NMP should be deleted from the figure. AILD and other terms should be spelled out in the figure legend. The definition of epitope spreading is inaccurate. Line 893 "differentially shape immune responses" is unclear. It should be paraphrased to explain that "trained immunity enhances autoimmunity, but innate tolerance suppresses it."
31. Figure 4A. "Antifungal Therapy MOA". What is "MOA"? "azoles" instead of "Triazoles." "pyrimidine analogs" instead of "5-FC converted to 5-FU." Echinocandins, "unmask beta glucan" instead of "inflammatory cytokine"
32. This reviewer recommends that Figure 4B be converted to a table format. In the new table, the first column will contain the therapy name, the second column will contain the antifungal effect, and the third column will contain the potential autoimmune induction.

Reply to Reviewers

We thank reviewers for their thoughtful and constructive comments. In this revision, we have made every effort to address all points raised and have revised the manuscript accordingly.

In this response document, reviewer comments are shown in **dark blue**, and our replies are in **black**. Revised sentences made to the main text are **dark green**.

We hope that the changes and clarifications we have provided satisfactorily address the concerns raised. Thank you again for your time and consideration.

Referee #1 (Remarks for Author):

This review article provides a broad overview of the bidirectional relationship between fungal infections and autoimmunity, highlighting the key mechanisms and therapeutic implications. This is a timely topic that will be of use and interest to a broad scientific audience. The organization by autoimmune and inflammatory conditions is particularly informative and different from other reviews on this topic. The figures are also detailed, though several of the words and icons are very small and difficult to see. My main critiques are outlined below.

We appreciate the reviewer's comments and feedback and have adopted changes to improve the readability of the figures.

(1) There's one issue not fully discussed in this review is the relationship between fungal infection and commensal fungi in the mycobiome. It's always good to discuss about the immune crosstalk with mycobiome in autoimmunity. But it's unclear whether the commensal-pathogen transition contributes to autoimmunity. Furthermore, how does the mycobiome dysbiosis contributes to fungal infection or vice versa?

We appreciate the reviewer's interest in furthering research that examines fungi as both commensal and pathogenic organisms. As suggested, we extensively revised the manuscript by creating new two subsections in the text. In the subsection "Shift of commensal to pathological fungi and its impacts on the host," we intended to lead the discussion to the plausible crosstalk between mycobiome and autoimmunity, which has yet to be fully established, based on previous studies. The subsection also discusses how the transition of fungal commensal to pathogen could happen. We also added another new subsection "Fungal Infections and Mycobiota Dysbiosis" that discusses how the mycobiome dysbiosis contributes to fungal infection and vice versa.

- Added as a new subsection "Shift of commensal to pathological fungi and its impacts on the host" in Section 3.a on Mycobiome.

Fungal species such as *C. albicans* and *Malassezia* spp. can shift from benign commensals to pathogenic organisms depending on host immune status and

environmental factors. This balance can be disrupted by antibiotics, antifungal treatments, or changes in the local microenvironment. As a result, fungi that normally reside harmlessly on mucosal and skin surfaces may contribute to disease when dysbiosis or immune dysfunction occurs—triggering inappropriate immune activation (Underhill & Iliiev, 2014) and potentially promoting autoimmunity. Although yet to be clearly demonstrated in the context of autoimmune disease, evidence suggests that prolonged antifungal treatments can alter gut fungal populations and exacerbate conditions such as colitis and allergic airway disease (Wheeler *et al*, 2016). These effects were mediated in part by gut-resident CX3CR1⁺ MNPs, which facilitate crosstalk between intestinal fungi and host immunity (Leonardi *et al*, 2018; Li *et al*, 2018). Although these studies suggested a role for commensal fungi in shaping immune responses, the causal mechanisms linking fungal dysbiosis to autoimmunity remain unclear. Future work should address whether the observed phenotypes are directly attributable to primarily to fungal pathogens or secondary immune alterations. Additionally, although antifungal treatments are necessary, the broader effects of antifungals on host immunity are not yet fully understood and may introduce variables that complicate the interpretation of fungal causality.

To date, much of the research on fungal dysbiosis has focused on mucosal inflammatory diseases, including IBD and lung allergy. Yet, given the influence of mycobiota on immune responses, it is plausible that fungal dysbiosis also plays a role in autoimmunity—particularly through its potential impact on adaptive immunity. Thus, it would be important to clarify the possible link between mycobiota and autoimmunity when considering future treatment strategies for individuals suffering from autoimmune diseases.

- Added as a new subsection “Mycobiota Dysbiosis and Fungal Infections” in Section 3.a on Mycobiome.

Not only does mycobiota dysbiosis result from fungal infections, but it also contributes to further infections. Perturbations of mycobiota—such as antibiotic use, immunosuppression, or epithelial barrier damage—can reduce host resistance and enable overgrowth or pathogenic transitions of fungi like *C. albicans* (Fan *et al*, 2015; Proctor *et al*, 2023). Expansion of such fungi can drive inflammation via secretion of virulence factors (e.g., candidalysin), activation of IL-17/IL-1 β pathways, and disruption of epithelial integrity, creating a feed-forward loop of dysbiosis and inflammation, particularly in diseases like inflammatory bowel disease (IBD) (Li *et al*, 2022; Moyes *et al*, 2016). Conversely, invasive fungal infection itself can reshape the mycobiome by outcompeting other species or altering the immune landscape, leading to long-term ecological shifts and increased susceptibility to secondary infections or chronic inflammation.

(2) In several studies discussed in this review, the claims (in cited studies) do not align with the presented data. The authors should discuss such shortcomings rather than solely highlighting the studies own claims. Shortcomings should be discussed, especially in such a nascent area of research.

We appreciate the reviewer’s important point. In this revised version, we carefully revisited the cited studies to ensure accurate interpretation. To strengthen the manuscript

and directly address this concern, we have introduced substantial new contents and added the following paragraphs and sentences in the revised text:

- **Fungal Dysbiosis in the Gut** – “Those shifts in mycobiota in patients raised intriguing possibilities for IBD treatments, but reverse causality cannot be ruled out. Further longitudinal studies are necessary to determine temporal relationships. Additionally, many studies have provided valuable insights into fungal community composition, but relatively few have incorporated functional analyses—such as metabolomic or transcriptomic profiling under reconstituted conditions with the fungal species of interest—which would be instrumental in clarifying the biological relevance of observed changes in the mycobiota.”
- **Shift of Commensal to Pathological Fungi and Its Impacts on the Host** -- “These studies elegantly described a role for commensal fungi in shaping immune responses, but the causal mechanisms linking fungal dysbiosis to autoimmunity remain unclear. Future work should address whether the observed phenotypes are directly attributable to primarily to fungal pathogens or secondary immune alterations. Additionally, although antifungal treatments are necessary, the broader effects of antifungals on host immunity are not yet fully understood and may introduce variables that complicate the interpretation of fungal causality.”
- **Risk and Challenges in Autoimmunity and IFIs** -- “there is increasing interest in the potential connection between fungal infections and the development of autoimmune diseases. Although this area of research is still in its early stages, emerging studies have suggested temporal associations between fungal infections and autoimmune onset. However, experimental models used to study fungal infection often rely on high inoculum doses that do not reflect natural exposure and may not precisely replicate the kinetics or localization of human disease—thereby limiting translational relevance. Furthermore, establishing a direct causal link remains challenging, as confounding factors such as pre-existing immune dysregulation or genetic susceptibility complicate interpretation. Addressing these limitations through improved modeling and mechanistic investigation will be essential to better understand the complex interplay between fungal pathogenesis and autoimmunity.”
- **Molecular Mimicry by Fungi** -- “Fungal mimicry of self-antigens presents an intriguing potential mechanism in disease pathogenesis, but its contribution has yet to be firmly established. Most existing studies rely on sequence homology-based predictions, with limited functional validation—such as demonstrating T or B cell cross-reactivity. Additional *in vivo* studies and immune profiling from patients will be critical to clarify the biological relevance of fungal mimicry in the development or progression of autoimmune diseases.”
- **Epitope spreading** -- “To date, direct evidence for such classical epitope spreading in the context of fungal infections remains lacking. For example, even though the immunological aftermath of fungal infection may create a permissive environment for autoimmunity, this should be distinguished from true epitope spreading as mechanistically defined. Future studies using antigen-specific tools and longitudinal tracking of immune responses will be essential to determine whether fungal infections can drive bona fide epitope spreading, or whether their role is limited to bystander inflammation and antigen release.”
- **Outcomes in Adaptive Immunity** – “a more comprehensive understanding of immunomodulatory fungal factors may be gained by incorporating physiologically relevant conditions, considering host-related variables such as genetic susceptibility and

microbiota composition, and further clarifying how cytokine responses, such as IL-17 induction, contribute to either protective immunity or pathological outcomes.

These additions aim to provide a more balanced and critical appraisal of the current literature, in line with the reviewer's helpful suggestion.

(3) While the review extensively discusses fungal pathogens and the mycobiome, recent advances emphasize that the effects of fungi on host immunity are modulated through interactions with other microbes, including bacteria, viruses, and archaea, within the microbial ecological network. For example, synergistic or antagonistic cross-kingdom interactions (e.g., fungal-bacterial metabolic crosstalk in the gut) can significantly influence immune responses. Also, fungal dysbiosis may exacerbate or mitigate autoimmune processes, largely related to the composition of the bacterial community. Incorporating these aspects would provide new insights into the fungal-autoimmune interactions within the broader ecological network of the integral microbiota.

We thank the reviewer for this insightful comment. We fully agree that cross-kingdom interactions represent an important and rapidly advancing area of research. While we were unable to expand on this topic in depth due to space constraints, we have now added a brief acknowledgment of this concept and cited several key references to highlight the relevance of fungal-bacterial metabolic crosstalk in modulating immune responses. The following sentences were added in the first paragraph of the Mycobiome section:

Although this review focuses on fungi, we recognize that fungal influences on host immunity often occur within the broader microbial ecological network. Notably, fungal–bacterial interactions, such as metabolic cross-feeding or competition in the gut, can shape immune and inflammatory outcomes, adding an important layer of complexity to mycobiome–host interactions. For more detailed discussions on cross-kingdom interactions in health and disease, we refer readers to several excellent reviews on the topic (Iliev & Leonardi, 2017; MacAlpine *et al*, 2023; Miyauchi *et al*, 2023; Shirliff *et al*, 2009).

(4) In addition, the review predominantly frames fungi as drivers of autoimmunity but does not sufficiently address their potential protective or regulatory roles. To provide a more balanced perspective, it is suggested to highlight the dual effects of commensal fungi. For instance, *Candida albicans* promotes Th17 responses that protect against pathogens but may also exacerbate inflammation in susceptible hosts.

This is a great point, and we have added the following sentences in the revised manuscript.

C. albicans is a well-characterized inducer of Th17 responses, which are critical for antifungal defense (Hernandez-Santos & Gaffen, 2012; LeibundGut-Landmann *et al*, 2007). However, Th17 responses can also contribute to immune-mediated pathology in genetically susceptible hosts or

under dysregulated inflammatory conditions, as exemplified in autoimmune disorders MS and RA. This dual role of Th17 immunity highlights the context-dependent nature of fungal–host interactions and the importance of immune balance in determining disease outcomes.

Although fungal infections typically elicit pro-inflammatory immune responses, β -glucan exposure can also regulate immune responses. For example, Dectin-1, a β -glucan receptor, mitigates EAE development via a neuroprotective mechanism through a CARD9-independent Dectin-1 signaling pathway (Deerhake *et al*, 2021). Intravenous injection of zymosan-depleted, a Dectin-1-specific agonist, delays the disease course (Deerhake *et al.*, 2021). β -glucan also delays T1D onset in murine models (Karumuthil-Melethil *et al*, 2014; Karumuthil-Melethil *et al*, 2008; Taylor & Vasu, 2021) by promoting the production of anti-inflammatory cytokine expression and increasing Treg frequency.

(5) Finally, clinical monitoring strategies for immunosuppressed patients are also important. The review acknowledges the increased risk of fungal infections in autoimmune patients undergoing immunomodulatory therapies but lacks discussion on actionable clinical guidance on how to manage this risk. For example, biomarker-based surveillance as well as personalized thresholds.

We appreciate the reviewer’s thoughtful comment regarding the importance of clinical monitoring strategies for immunosuppressed patients at risk of invasive fungal infections (IFIs), particularly in the context of autoimmune diseases treated with immunomodulatory therapies. Indeed, recognizing and managing IFIs in this population remains a critical challenge in clinical practice. While our review focuses primarily on mechanistic and immunological insights, we agree that bridging these findings to actionable clinical strategies is essential.

We have added a brief statement below to the revised text to emphasize this translational gap and highlight the need for validated, context-specific surveillance strategies that could help identify high-risk individuals and guide antifungal prophylaxis or early intervention.

Although recent advances have explored blood biomarkers for IFI detection (including detection of fungi by PCR or identification of antigens), the gold-standard continues to rely on clinical suspicion coupled with evidence from microscopy or serial blood cultures (Cornely *et al*, 2025; Galmiche *et al*, 2023; Taynton *et al*, 2022). Though essential, these traditional methods are often time-consuming, require specialized expertise, and may delay appropriate treatment. Broader adoption, refinement, and validation of novel diagnostic tools will be critical to enable more timely and accurate detection of fungal infections, particularly in vulnerable patient populations.

Notably, despite the advances in understanding host-fungal interactions, clinical recognition of IFIs in patients with immune-mediated diseases remains challenging. In addition to conventional diagnostics, molecular and biomarker-based assays (*e.g.*, galactomannan, β -glucans, and fungal genetic taxonomy) are under development and show promise for earlier detection (Fang *et al*, 2023). Validation of these approaches in immunomodulated autoimmune cohorts and developing personalized thresholds to guide actionable surveillance will be important next steps.

(6) Figures: Provide a list of fungal species, either commensal or pathogenic, in different autoimmune diseases.

Thank you for your suggestion. We made added a new table (**Table 2**), a list of fungal species and their associations with various autoimmune diseases

(7) Moreover, I would suggest a figure describing the triple interaction of "fungal infection - mycobiome - autoimmunity".

We appreciate the reviewer's suggestion to include a figure illustrating the 3-way interaction. As a part of our responses, we added the new subsection "Fungal Dysbiosis and Fungal Infections" to discuss how dysbiosis may contribute to fungal infection and, conversely, how fungal infection may alter the composition and function of the mycobiome. However, when it comes to the balanced description of the 3-way connection, the current evidence remains limited and largely speculative particularly on the end of autoimmunity. Given the early stage of this research area and the lack of well-defined mechanistic studies linking all three elements, we believe it is more appropriate to address this triad conceptually in the text rather than in a figure, with which the nuance is difficult to convey.

Addressing these points would significantly enhance the manuscript by incorporating fungal-autoimmune interactions within a multispecies ecological model, providing a more balanced view of fungal roles, clarifying the therapeutic risks associated with emerging biologic therapies, and offering practical, personalized clinical recommendations. The revisions would better align the review with current trends in microbiome research and precision medicine, ultimately increasing its impact and clinical relevance.

We greatly appreciate the reviewer's comments. We have made substantial additions and edits as suggested, and we hope that the revised manuscript is now significantly improved.

Referee #2 (Remarks for Author):

In this review article, the authors described the physiological (beneficial) and pathological (detrimental) roles of fungi and antifungal immune responses. The review summarized the relationships between representative fungi and major immune cells, discussing the potential risk and mechanism(s) of induction of immune-mediated diseases by fungal infections; these topics have rarely been reviewed by others. Thus, this reviewer recommends the manuscript for publication. In this manuscript, however, several issues need to be addressed: for example, the inaccurate definition of several terms and the overemphasis of several findings that were only loosely associated with fungal infections.

We sincerely thank the reviewer for their thoughtful and encouraging assessment of our manuscript. We are especially grateful for the recognition that our review addresses the underexplored intersection between fungal infections and immune-mediated diseases. In response to the reviewer's helpful suggestions, we have carefully revised the manuscript to clarify the definitions of key terms and to adjust language in places where certain findings may have been overinterpreted in the context of fungal infection. We believe these revisions have strengthened the clarity and balance of the review.

Major concerns

(1) The authors used the present tense in most sentences in this manuscript. This is incorrect; most experimental and clinical findings should be described using past or present perfect tense. For example, in section 4, most sentences were based on experimental and clinical findings, thus should be described in the past tense (if the finding was based on a few manuscripts) or the present perfect tense (if the finding was based on several manuscripts in the past 10 years or so). Present tense usage gives the reader the wrong impression that the description was established as a permanent truth. Other examples of misusages of the present tense are as follows: on line 49, "linked" instead of "links". All sentences on lines 202-212.

We agree that the past or present perfect tense is more appropriate when describing experimental and clinical findings, especially to avoid implying undue generalization or permanence. In response, we have carefully revised the manuscript, particularly Section 4 and line 49 to use the appropriate tense based on the nature and strength of the supporting evidence. Regarding the previous lines 202–212 (originally under the “Fungal Dysbiosis and Gut Inflammation” subsection), those sentences no longer exist in the revised version, as this section was substantially edited in response to Reviewer 1–(2).

In the revised text, we have carefully attended to the use of appropriate verb tenses to ensure clarity and accuracy. We believe these changes strengthen the overall presentation of the manuscript.

(2-a) The usage of several terms, including "autoimmune disease" and "autoinflammatory disease," was inaccurate (e.g., line 203). As the authors mentioned, inflammatory bowel disease (IBD) is not categorized as an autoimmune disease. There are other diseases in this manuscript, which are not categorized as autoimmune diseases: spondyloarthritis, celiac disease (line 231), and animal models on lines 220-222.

We thank the reviewer for catching the wording and have carefully re-evaluated the use of the terms “autoimmune” throughout the manuscript. We agree that clarity and precision are critical in categorizing immune-mediated diseases. For example, to ensure accurate and consistent use of disease classifications, we changed the structure of the subsection “Fungal Dysbiosis in Host Pathology of Distant Tissues” by focusing on classical autoimmune diseases (MS, SLE, RA) in the first paragraph. Then, other diseases---which are not considered to be autoimmune but induced by inflammation and other immune-related mechanisms--are discussed in the second paragraph.

(2-b) In general, immune cells in acquired immunity play a major role in autoimmune diseases; activation of innate immune cells causes (hereditary) autoinflammatory diseases (e.g., on line 265, "autoinflammatory disease development" should be used, instead of "autoimmune disease development." Here, it should be noted that an elevated innate immunity can enhance, but not prime, autoimmune responses.).

Thank you for bringing out the issue. We tried to be careful with our wording. Yet, for the sentence previously located at line 265 (“*These studies suggest fungal infection could elevate systemic innate immune inflammation, potentially contributing to autoimmune disease development.*”), we would like to clarify that innate immunity was not described to “prime” autoimmune responses. Rather, we noted that inflammation triggered by innate immune activation could “contribute” to autoimmune disease development. For example, IL-1 β produced by innate immune cells contributes—though does not prime—Th17 responses, a mechanism that has been particularly well characterized in the context of autoimmune models such as experimental autoimmune encephalomyelitis (EAE).

(2-c) The authors misused "autoinflammatory" as "autoimmune" in the text. This reviewer recommends using the term "immune-mediated" instead of "autoimmune" in most sentences in this article, including the title (e.g., lines 143, 164).

We fully understand the reviewer’s point regarding the careful use of the term “*autoimmune.*” In response, we have thoroughly reviewed the manuscript and revised the language to ensure the term is used appropriately and precisely.

Sentence previously located at line 143 stated: “*Th1 and Th17 cell responses also contribute to autoimmune diseases such as MS, RA, and CD.*” MS and RA are classical autoimmune diseases, we preferred to use “autoimmune,” but CD was removed because it is not an autoimmune disease. We apologize for the oversight.

Sentence previously locate at line 164 stated: “*Early inflammation helps contain invading pathogens, but uncontrolled or prolonged immune responses can hinder pathogen clearance and trigger off-target tissue damage, contributing to autoimmunity.*” We believe that the use of the term “*autoimmunity*” in this context is appropriate. As noted in our response to Comment 2-b, we refer to early inflammation “*contributing*” to autoimmunity—not “*priming*” it. Indeed, early inflammation does play a contributory role in the development of autoimmunity; for example, this concept underlies the use of complete Freund’s adjuvant (CFA) in autoimmune disease model inductions such as experimental autoimmune encephalomyelitis (EAE) and Collagen-Induced Arthritis (CIA).

Given that the manuscript specifically addresses autoimmune diseases, we prefer to keep the defined word “autoimmune” for diseases, such as MS, SLE, RA, and T1D. The reviewer wrote “using the term *immune-mediated* instead of *autoimmune* in most sentences in this article.” However, using “immune-mediated” for most sentences in this article does not sound very realistic because of numerous descriptions of autoimmunity in this manuscript. Thus, not using *autoimmunity* in most sentences would risk obscuring autoimmunity. Importantly, our primary intention of this reviewer article is to connect

fungal infections and autoimmunity. Therefore, regarding the title, we respectfully request to retain it as is, as this review is specifically focused on autoimmunity. Although we occasionally mention non-autoimmune inflammatory diseases, these references are intended to extend discussions and to illustrate deductive mechanisms that may help to understand autoimmune processes during fungal infections. Replacing “autoimmunity” with “immune-mediated,” which implies a significantly broader scope, would misrepresent the manuscript’s central focus and would require a substantial revision of the entire content.

We appreciate the reviewer’s suggestion and have taken care throughout the revised manuscript to use terminology more precisely.

(2-d) The term "immune-mediated diseases" includes autoimmune diseases, autoinflammatory diseases, and other immune-mediated diseases with unknown etiology or induced by microbial infections. Thus, the term "immune-mediated disease" is more appropriate than "autoimmune diseases" in this article.

We appreciate the important point raised by the reviewer. This comment included the wording “autoinflammatory diseases,” but this manuscript contains no discussion of autoinflammatory diseases (such as Familial Mediterranean Fever, TNF Receptor–Associated Periodic Syndrome, Cryopyrin-Associated Periodic Syndromes). Therefore, using the broader term “*immune-mediated diseases*”—which encompasses autoinflammation—may be unnecessarily expansive, although we tried to use the words in the text when reasonable. To maintain clarity and focus, we have taken a consistent approach throughout the manuscript to distinguish carefully between autoimmune and non-autoimmune mechanisms and to use the term “*autoimmunity*” only when specifically appropriate.

(3) In this manuscript, the two substantial pieces of evidence linking fungal infections and autoimmune diseases are found in ASCA and beta-glucan, although associations of most other findings with autoimmune diseases cited in this manuscript were not strong, with little supportive evidence. The findings of ASCA and beta-glucan were described in several sections, often redundantly (e.g., lines 95, 156, 216, 334). Thus, this reviewer recommends setting up two new paragraphs, each describing the findings of ASCA and beta-glucan with more explanations as the most strong evidence linking fungal infections and autoimmunity.

As suggested, we generated a new subsection (Outcomes in Adaptive Immunity), where ASCA and β -glucans are discussed. The new section reads as follows:

Numerous innate immune receptors sense fungi. Yet, Dectin-1 signaling in myeloid cell types on shape adaptive immunity have been extensively studied in fungal infections. Ligating Dectin-1 with β -glucans promotes an antifungal immune response and protects mammalian hosts via triggering reactive oxygen species (ROS), phagocytosis, and proinflammatory cytokine expression (Deerhake & Shinohara, 2021) at the levels of innate immunity, and ultimately promotes Th17 responses (LeibundGut-Landmann *et al.*, 2007). *C. albicans* is a well-characterized inducer of Th17 responses, which are critical for antifungal defense (Hernandez-Santos & Gaffen, 2012; LeibundGut-Landmann *et al.*, 2007). However, Th17 responses can also

contribute to immune-mediated pathology in genetically susceptible hosts or under dysregulated inflammatory conditions, as exemplified in autoimmune disorders such as MS and RA. This dual role of Th17 immunity highlights the context-dependent nature of fungal–host interactions and the importance of immune balance in determining disease outcomes.

Although fungal infections typically elicit pro-inflammatory immune responses, β -glucan exposure can also regulate immune responses. For example, Dectin-1, a β -glucan receptor, mitigates EAE development via a neuroprotective mechanism through a CARD9-independent Dectin-1 signaling pathway (Deerhake *et al.*, 2021). Intravenous injection of zymosan-depleted, a Dectin-1-specific agonist, delays the disease course (Deerhake *et al.*, 2021). β -glucan also delays T1D onset in murine models (Karumuthil-Melethil *et al.*, 2014; Karumuthil-Melethil *et al.*, 2008; Taylor & Vasu, 2021) by promoting the production of anti-inflammatory cytokine expression and increasing Treg frequency. Research is ongoing to understand how the β -glucans/Dectin-1 axis can be either proinflammatory or regulatory. At least, a current explanation is that functional outcomes depend on distinct signaling pathways downstream of Dectin-1 (Deerhake *et al.*, 2021; Deerhake & Shinohara, 2021).

Our discussion also encompasses anti-fungal antibodies: ASCAs serve as serological markers for CD (Main *et al.*, 1988) and are found in patients with autoimmune disorders, such as SLE (Dai *et al.*, 2009a; Mankai *et al.*, 2013), T1D (Sakly *et al.*, 2010), and RA (Dai *et al.*, 2009b). Although “ASCAs” were named after *S. cerevisiae*, the fungus was very rarely isolated from stools and mouth swabs from patients, including those with CD (Sendid *et al.*, 2024). Instead, major fungal sources of ASCA epitopes are now considered to be *Candida* spp. (Muller *et al.*, 2010; Schaffer *et al.*, 2007; Sendid *et al.*, 2024). Although the precise mechanism remains under investigation, ASCAs may contribute to inflammation, particularly in the gut, through multiple interacting factors.

These studies highlight how contextual cues in adaptive immunity shape immune responses to fungi in autoimmune diseases. Yet, a more comprehensive understanding of immunomodulatory fungal factors may be gained by incorporating physiologically relevant conditions, considering host-related variables such as genetic susceptibility and microbiota composition, and further clarifying how cytokine responses, such as IL-17 induction, contribute to either protective immunity or pathological outcomes.

(4) Section 3b) Molecular mimicry. In bacteria and virus infections, antibodies, not T cells, have been reported to recognize cross-reactive epitopes by molecular mimicry, for example, anti-glycolipid antibodies and *Campylobacter jejuni* in Guillan Barre syndrome or brain antigens and EB virus in multiple sclerosis. On the other hand, in fungal infections, T cells seemed to recognize cross-reactive epitopes, according to this manuscript. Are there any examples of cross-reactive antibodies in fungal infections? If not, it is good to speculate/discuss why cross-reactive T cells, but not antibodies, have been reported in fungal infections, unlike bacteria or viruses (e.g., differences in complement activation, role of T cells, plasma membrane structure, etc).

Because of the previous statement (*Unlike bacteria and viruses, eukaryotic fungi share greater structural homology with human proteins, increasing the risk of cross-reactivity*) combined with our discussion of TCR sequencing, we may have inadvertently given the impression that fungal molecular mimicry predominantly involves T cell–mediated mechanisms. This was not our intention. In reality, only a limited number of studies have implicated fungal molecular mimicry and did not go beyond identifying sequences homologous to fungal epitopes, which were predicted to bind TCRs from

patients with autoimmune diseases. To avoid this confusion, we have revised the subsection as follows:

Molecular mimicry occurs when microbial antigens resemble host molecules, potentially triggering autoreactive immune responses (**Fig. 1B**). In the context of fungal pathogens, evidence for molecular mimicry remains limited. So far, studies have primarily identified fungal epitopes through T cell receptor (TCR) sequencing data from patients with autoimmune disorders (Grogan *et al*, 1999; Repac *et al*, 2021; Whalley *et al*, 2020). Fungal mimicry of self-antigens presents an intriguing potential mechanism in disease pathogenesis, but its contribution has yet to be firmly established. Most existing studies rely on sequence homology-based predictions, with limited functional validation—such as demonstrating T or B cell cross-reactivity. Additional *in vivo* studies and immune profiling from patients will be critical to clarify the biological relevance of fungal mimicry in the development or progression of autoimmune diseases.

(5) Section 3c) epitope spreading. Although the authors cited EAE as an example of epitope spreading of EAE, a more appropriate citation should be epitope spreading in microbial infections. For instance, in reference 96, Stephen Miller's group described epitope spreading in a viral model of multiple sclerosis, Theiler's murine encephalomyelitis virus (TMEV) infection in mice. In the TMEV model, T cells derived from TMEV-infected mice recognized TMEV antigens, initially. However, inflammation in the spinal cord resulted in the release of myelin antigen, leading to epitope spreading to PLP139-151, which TCR recognizes. Later, epitopes further spread to PLP171-191, followed by MBP84-104, and lastly to MOG96-102. In TMEV infection, although direct virus infection also causes myelin destruction, it is not regarded as the primary cause of epitope spreading. In microbial infections, the definition of epitope spreading is the changes of the immune epitope from microbial epitope to host epitope; this has been described in other diseases, including rheumatic disease and experimental HSV infection. Thus, in fungal infection, the term "epitope spreading" should be used only when fungal epitopes recognized by TCR or BCR spread to host epitopes recognized by TCR or BCR. Induction of autoimmune cells by the release of host proteins following direct fungal infection in the organ should not be called epitope spreading. Thus, the entire paragraph of section 3c) should be rewritten.

We agree with the reviewer for pointing us to the seminal studies on epitope spreading in microbial infections, particularly the Theiler's murine encephalomyelitis virus (TMEV) model. Now, the revision mentions the model. We also appreciate the distinction between the classical definition of epitope spreading—the progression of immune responses from microbial epitopes to self-epitopes—as opposed to immune activation resulting from host tissue damage alone. In the revised manuscript, we have rewritten Section 3c to reflect this distinction. Specifically, we now clarify that, although fungal infections may lead to tissue injury, current evidence does not clearly demonstrate classical epitope spreading driven by initial recognition of fungal antigens. Please find the re-written Section 3c below:

Epitope spreading is an immune phenomenon where an immune response initially targets a specific epitope of an antigen and later broadens to recognize additional epitopes on the same or different proteins (**Fig. 1B**). It is particularly well-studied in animal models of autoimmune

demyelination. In experimental autoimmune encephalomyelitis (EAE), an immune response initially targeting a myelin protein (e.g., myelin oligodendrocyte glycoprotein, MOG) later spreaded to other myelin components (MBP, PLP) (Getts *et al*, 2013). In the Theiler's murine encephalomyelitis virus (TMEV) model, the initial virus responses to CNS infection led to host auto-reactivity to myelin epitopes, host tissue destruction, and myelin antigen presentation (Miller *et al*, 1997). To date, direct evidence for such classical epitope spreading in the context of fungal infections remains lacking. For example, even though the immunological aftermath of fungal infection may create a permissive environment for autoimmunity, this should be distinguished from true epitope spreading as mechanistically defined. Future studies using antigen-specific tools and longitudinal tracking of immune responses will be essential to determine whether fungal infections can drive *bona fide* epitope spreading, or whether their role is limited to bystander inflammation and antigen release.

Minor concerns

1. The authors sometimes did not cite the key references that should be required for understanding the manuscript's contents. In Section 1, there were no single references. At least, a sentence on lines 50-51 needs references. The following were lines that need references: lines 75, 158.

Section 1 was intended as a general introduction to provide context before delving into more detailed content in Section 2 and beyond; therefore, no citations were included in this section. However, we agree that the previous statement “*For instance, Candida albicans, Cryptococcus neoformans, and Aspergillus fumigatus have been associated with autoimmune conditions*” (previously in lines 50-51) would have benefited from appropriate referencing. As there are numerous relevant studies—many of which are cited in the subsequent sections—we felt it was more appropriate to remove the sentence from the introduction to maintain clarity and avoid redundancy.

We included the reference for the previous line 75 description about WHO FPPL (WHO, 2020). Previous line 158 “*Memory B cells may trigger autoimmunity, if fungal antigens cross-react with self epitopes.*” This sentence is our speculation. To clarify, we added “*It is possible that...*”

2. Line 85, "people" instead of "those"

Edited as suggested. (Currently Line 84)

3. Line 88, "MS patients" instead of "patients"

Edited as suggested. (Currently Line 86)

4. Line 89, "will be discussed" instead of "are discussed "

Edited as suggested. (Currently Line 87)

5. Line 93, "Crohn's disease" instead of "Crohn's Disease"

Edited as suggested. (Currently Line 89)

6. Line 93, "SLE" should be spelled out here, instead of lines 226-227. The following terms should be also spelled out when the terms were used for the first time: IL-23 (line 108), TNF (line 108), Th (line 109), ROS (line 111), IFN (line 128), UC (line 201)

The following words have been spelled out as suggested:

SLE (Currently Line 91)

IL-23 (Currently Line 104)

TNF (Currently Line 104)

Th (Currently Line 105)

ROS (Currently Line 107)

IFN (Currently Line 126)

UC (Currently Line 218)

7. Line 102, delete "but not limited to"

Deleted as suggested.

8. The ambiguous word "while" is overused; in most sentences, it can be replaced with the more specific word "although," for example, on lines 122, 155, and 172

We replaced "while" to other wording. Now there is no single usage of "while" in the document.

9. Line 126, "(NK) cells" instead of "(NK)

Edited as suggested. (Currently Line 123)

10. Line 146, "Th" instead of "T helper"

Edited as suggested. (Currently Line 141)

11. Line 153, "cellular cytotoxicity" instead of "cell toxicity"

Edited as suggested. (Currently Line 152)

12. Line 173, "human gut" instead of "gut"

Edited as suggested. (Currently Line 170)

13. Line 178, "correlative" is unclear. This sentence needs to be paraphrased.

The previous sentences, including the one with "correlative," have been rephrased to address Reviewer #1's comment (3). The current version no longer includes the description containing the term "correlative."

14. Line 184, "MNPs" need more explanation, such as their phenotype and origin.

Although the phenotypes and origins of these cells are certainly intriguing, the topic delves into detailed aspects of myeloid cell biology. Due to space constraints, we have opted not to expand on the mononuclear phagocytes (MNPs) biology *per se*, and instead added a brief explanation to help readers understand the term. We also added MNPs to the Glossary.

15. Line 196, "inflammatory bowel disease" instead of "Inflammatory Bowel Disease"

We have incorporated the change, as suggested (Line 204).

16. Line 203, "IBD" instead of "autoinflammatory disease"

The original sentence was entirely rewritten instead.

17. Line 212, "IBD" instead of "inflammatory disease"

The original sentence was entirely rewritten instead.

18. Line 216, there are several "autoimmune liver disease." This needs more specification.

Revision now describes:

Immune-mediated liver diseases (ILD), including primary biliary cholangitis, autoimmune hepatitis, and primary sclerosing cholangitis...

19. Line 230: Since there are several types of spondyloarthropathy, a more specific description or explanation is required here.

We updated the description as follows:

Patients with ankylosing spondylitis, not a classical autoimmune disease but immune-mediated...

20. Line 307, "compared with" instead of "compared to"

Edited as suggested. (Currently Line 380)

21. Line 365, "enhance inflammatory genes in immune cells" needs more explanations.

We have now specified *Tnf*.

22. Line 368, "its immunostimulatory effects" needs more explanations.

We update the description as follows:

Direct evidence linking 5-FU to autoimmunity is still unknown, but its immunostimulatory effects, such as cytotoxicity on myeloid-derived suppressor cells (MDSCs), were reported in a cancer setting (Vincent *et al*, 2010)

23. Lines 371-373. This sentence does not make sense; the sentence should be rewritten or deleted.

The original description was “*While effective against IFIs, antifungal therapies may induce immune stimulation and DAMP release, potentially driving bystander activation and improper self-antigen targeting, which may trigger or exacerbate autoimmunity.*”

We rephrased the sentence as follows:

Antifungal therapies, though effective against IFIs, can also provoke immune activation and DAMP release, which may inadvertently trigger bystander immune responses and self-antigen recognition, contributing to the onset or aggravation of autoimmunity.

24. Line 387, "immunotoxicity risks" need more explanations.

We rephrased the sentence. In particular, instead of “immunotoxicity risks,” we used wording “immune-related adverse events.”

25. Line 388 and figure 4, "Monoclonal antibodies" need explanations. Are they antifungal antibodies or some other antibodies?

The wording has now been revised to read “Anti-fungal monoclonal antibodies.”

26. Line 388, "knowledge gaps". Are there any reports of induction of autoimmunity in the treatment described here?

We have described the existing knowledge gaps regarding the potential impacts of antifungal antibodies and vaccines on autoimmunity. Although there are indirect implications—such as the capacity of antifungal antibodies to induce proinflammatory responses, which could theoretically exacerbate autoimmunity—these concerns remain speculative due to the absence of solid mechanistic evidence. In this revision, we have added further clarification emphasizing the need for studies that assess endpoints of autoimmune tissue damage following these treatments, in order to better understand the possible off-target effects of biologic therapeutics used for invasive fungal disease.

Anti-fungal monoclonal antibodies and vaccines have been explored as strategies to treat and prevent invasive fungal infections (IFIs), showing promising results in preclinical studies (Bromuro *et al*, 2010; Bugli *et al*, 2013; Pahl *et al*, 2006). However, limitations in our understanding of their functional and mechanistic links to autoimmunity remain a key barrier to clinical translation.

27. Lines 400-401. This sentence needs to be rewritten. The authors need to discuss more specifically how G-CSF or CAR-T cell treatment potentially causes autoimmune diseases.

We appreciate the reviewer's request for more information. We have now incorporated explanation as follows:

Both granulocytes (as major producers of pro-inflammatory cytokines, proteases, and ROS) and CAR-T cells (through the production of pro-inflammatory cytokines) have the potential to exacerbate systemic inflammation that may activate auto-reactive T and B cells.

28. The conclusion section is too long. It should be written in one paragraph with a more explicit message.

As suggested, the conclusion section was shortened in one paragraph.

29. Figure 1. The following 1) to 4) were not explained in the text. Thus, the authors should delete the phrases or include the explanations in the text. 1) Neutrophil: contribution of chronic inflammation; promotes autoimmunity in leukocytes. 2) Dendritic cell: pathogen containment by granuloma formation; May promote immune exhaustion. 3) T cell: granuloma formation. 4) B cell: formation of ectopic germinal centers. In addition, ROS, NETs, PRRs, Th should be spelled out in figure legend.

As suggested by the editor, Figure 1 has been converted to Table 2. To reply to this comment, Table 2 now includes information, either mentioned in the main text or generally well-accepted. That said, we removed the items that were mentioned by this reviewer. As for the abbreviations, the information is now shown as a Table and cannot add a legend. Thus, we confirmed the abbreviations are spelled out in the main text.

30. Figure 2. Myeloid NMP was not explained well in the text. Thus, more explanations on NMP should be included in the text, or NMP should be deleted from the figure. AILD and other terms should be spelled out in the figure legend. The definition of epitope spreading is inaccurate. Line

893 "differentially shape immune responses" is unclear. It should be paraphrased to explain that "trained immunity enhances autoimmunity, but innate tolerance suppresses it."

- For MNPs, please see our reply in the Minor concern #14.
- We spelled out the abbreviations and revised the definition of epitope spreading.
- The Fig. 2 legend has been entirely re-written.

31. Figure 4A. "Antifungal Therapy MOA". What is "MOA"? "azoles" instead of "Triazoles." "pyrimidine analogs" instead of "5-FC converted to 5-FU." Echinocandins, "unmask beta glucan" instead of "inflammatory cytokine"

- We removed "MOA" (mechanism of action) from the figure.
- We replaced "Triazoles" to "Azoles."
- We replaced "5-FC converted to 5-FU" to "pyrimidine analogs."
- We respectfully request to retain the term "inflammatory cytokines" in the figure, as it cannot be logically replaced by "echinocandins" or "unmask beta glucan." The intention in the figure context is to convey an increase in inflammatory cytokine production, which is a downstream immune consequence rather than a direct action of the antifungal agent.

32. This reviewer recommends that Figure 4B be converted to a table format. In the new table, the first column will contain the therapy name, the second column will contain the antifungal effect, and the third column will contain the potential autoimmune induction.

Thank you for the suggestion. We felt that including visual representations would be helpful, especially for readers who are new to the field, such as trainees. Also, we were hesitant to create a table listing "potential autoimmune induction," as this format might unintentionally convey that such effects are scientifically established rather than speculative. To avoid this potential misunderstanding, we chose to focus the visual information on the therapies currently being evaluated, without implying potential autoimmune outcomes.

References

Bromuro C, Romano M, Chiani P, Berti F, Tontini M, Proietti D, Mori E, Torosantucci A, Costantino P, Rappuoli R *et al* (2010) Beta-glucan-CRM197 conjugates as candidates antifungal vaccines. *Vaccine* 28: 2615-2623

Bugli F, Cacaci M, Martini C, Torelli R, Posteraro B, Sanguinetti M, Paroni Sterbini F (2013) Human monoclonal antibody-based therapy in the treatment of invasive candidiasis. *Clin Dev Immunol* 2013: 403121

Cornely OA, Sprute R, Bassetti M, Chen SC, Groll AH, Kurzai O, Lass-Flörl C, Ostrosky-Zeichner L, Rautemaa-Richardson R, Revathi G *et al* (2025) Global guideline for the diagnosis and management of

candidiasis: an initiative of the ECMM in cooperation with ISHAM and ASM. *Lancet Infect Dis* 25: e280-e293

Dai H, Li Z, Zhang Y, Lv P, Gao XM (2009a) Elevated levels of serum antibodies against *Saccharomyces cerevisiae* mannan in patients with systemic lupus erythematosus. *Lupus* 18: 1087-1090

Dai H, Li Z, Zhang Y, Lv P, Gao XM (2009b) Elevated levels of serum IgA against *Saccharomyces cerevisiae* mannan in patients with rheumatoid arthritis. *Cell Mol Immunol* 6: 361-366

Deerhake ME, Danzaki K, Inoue M, Cardakli ED, Nonaka T, Aggarwal N, Barclay WE, Ji RR, Shinohara ML (2021) Dectin-1 limits autoimmune neuroinflammation and promotes myeloid cell-astrocyte crosstalk via Card9-independent expression of Oncostatin M. *Immunity* 54: 484-498 e488

Deerhake ME, Shinohara ML (2021) Emerging roles of Dectin-1 in noninfectious settings and in the CNS. *Trends Immunol* 42: 891-903

Fan D, Coughlin LA, Neubauer MM, Kim J, Kim MS, Zhan X, Simms-Waldrip TR, Xie Y, Hooper LV, Koh AY (2015) Activation of HIF-1alpha and LL-37 by commensal bacteria inhibits *Candida albicans* colonization. *Nat Med* 21: 808-814

Fang W, Wu J, Cheng M, Zhu X, Du M, Chen C, Liao W, Zhi K, Pan W (2023) Diagnosis of invasive fungal infections: challenges and recent developments. *J Biomed Sci* 30: 42

Galmiche S, Thoreau B, Bretagne S, Alanio A, Paugam A, Letscher-Bru V, Cassaing S, Gangneux JP, Guegan H, Favennec L *et al* (2023) Invasive fungal diseases in patients with autoimmune diseases: a case series from the French RESSIF network. *RMD Open* 9

Getts DR, Chastain EM, Terry RL, Miller SD (2013) Virus infection, antiviral immunity, and autoimmunity. *Immunol Rev* 255: 197-209

Grogan JL, Kramer A, Nogai A, Dong L, Ohde M, Schneider-Mergener J, Kamradt T (1999) Cross-reactivity of myelin basic protein-specific T cells with multiple microbial peptides: experimental autoimmune encephalomyelitis induction in TCR transgenic mice. *J Immunol* 163: 3764-3770

Hernandez-Santos N, Gaffen SL (2012) Th17 cells in immunity to *Candida albicans*. *Cell Host Microbe* 11: 425-435

Iliev ID, Leonardi I (2017) Fungal dysbiosis: immunity and interactions at mucosal barriers. *Nat Rev Immunol* 17: 635-646

Karumuthil-Meethil S, Gudi R, Johnson BM, Perez N, Vasu C (2014) Fungal beta-glucan, a Dectin-1 ligand, promotes protection from type 1 diabetes by inducing regulatory innate immune response. *J Immunol* 193: 3308-3321

Karumuthil-Meethil S, Perez N, Li R, Vasu C (2008) Induction of innate immune response through TLR2 and dectin 1 prevents type 1 diabetes. *J Immunol* 181: 8323-8334

LeibundGut-Landmann S, Gross O, Robinson MJ, Osorio F, Slack EC, Tsoni SV, Schweighoffer E, Tybulewicz V, Brown GD, Ruland J *et al* (2007) Syk- and CARD9-dependent coupling of innate immunity to the induction of T helper cells that produce interleukin 17. *Nat Immunol* 8: 630-638

Leonardi I, Li X, Semon A, Li D, Doron I, Putzel G, Bar A, Prieto D, Rescigno M, McGovern DPB *et al* (2018) CX3CR1(+) mononuclear phagocytes control immunity to intestinal fungi. *Science* 359: 232-236

Li X, Leonardi I, Semon A, Doron I, Gao IH, Putzel GG, Kim Y, Kabata H, Artis D, Fiers WD *et al* (2018) Response to Fungal Dysbiosis by Gut-Resident CX3CR1(+) Mononuclear Phagocytes Aggravates Allergic Airway Disease. *Cell Host Microbe* 24: 847-856 e844

Li XV, Leonardi I, Putzel GG, Semon A, Fiers WD, Kusakabe T, Lin WY, Gao IH, Doron I, Gutierrez-Guerrero A *et al* (2022) Immune regulation by fungal strain diversity in inflammatory bowel disease. *Nature* 603: 672-678

MacAlpine J, Robbins N, Cowen LE (2023) Bacterial-fungal interactions and their impact on microbial pathogenesis. *Mol Ecol* 32: 2565-2581

Main J, McKenzie H, Yeaman GR, Kerr MA, Robson D, Pennington CR, Parratt D (1988) Antibody to *Saccharomyces cerevisiae* (bakers' yeast) in Crohn's disease. *BMJ* 297: 1105-1106

Mankai A, Sakly W, Thabet Y, Achour A, Manoubi W, Ghedira I (2013) Anti-*Saccharomyces cerevisiae* antibodies in patients with systemic lupus erythematosus. *Rheumatol Int* 33: 665-669

Miller SD, Vanderlugt CL, Begolka WS, Pao W, Yauch RL, Neville KL, Katz-Levy Y, Carrizosa A, Kim BS (1997) Persistent infection with Theiler's virus leads to CNS autoimmunity via epitope spreading. *Nat Med* 3: 1133-1136

Miyauchi E, Shimokawa C, Steimle A, Desai MS, Ohno H (2023) The impact of the gut microbiome on extra-intestinal autoimmune diseases. *Nat Rev Immunol* 23: 9-23

Moyes DL, Wilson D, Richardson JP, Mogavero S, Tang SX, Wernecke J, Hofs S, Gratacap RL, Robbins J, Runglall M *et al* (2016) Candidalysin is a fungal peptide toxin critical for mucosal infection. *Nature* 532: 64-68

Muller S, Schaffer T, Flogerzi B, Seibold-Schmid B, Schnider J, Takahashi K, Darfeuille-Michaud A, Vazeille E, Schoepfer AM, Seibold F (2010) Mannan-binding lectin deficiency results in unusual antibody production and excessive experimental colitis in response to mannose-expressing mild gut pathogens. *Gut* 59: 1493-1500

Pachl J, Svoboda P, Jacobs F, Vandewoude K, van der Hoven B, Spronk P, Masterson G, Malbrain M, Aoun M, Garbino J *et al* (2006) A randomized, blinded, multicenter trial of lipid-associated amphotericin B alone versus in combination with an antibody-based inhibitor of heat shock protein 90 in patients with invasive candidiasis. *Clin Infect Dis* 42: 1404-1413

Proctor DM, Drummond RA, Lionakis MS, Segre JA (2023) One population, multiple lifestyles: Commensalism and pathogenesis in the human mycobiome. *Cell Host Microbe* 31: 539-553

Repac J, Mandic M, Lunic T, Bozic B, Bozic Nedeljkovic B (2021) Mining the capacity of human-associated microorganisms to trigger rheumatoid arthritis-A systematic immunoinformatics analysis of T cell epitopes. *PLoS One* 16: e0253918

Sakly W, Mankai A, Sakly N, Thabet Y, Achour A, Ghedira-Besbes L, Jeddi M, Ghedira I (2010) Anti-*Saccharomyces cerevisiae* antibodies are frequent in type 1 diabetes. *Endocr Pathol* 21: 108-114

Schaffer T, Muller S, Flogerzi B, Seibold-Schmid B, Schoepfer AM, Seibold F (2007) Anti-*Saccharomyces cerevisiae* mannan antibodies (ASCA) of Crohn's patients crossreact with mannan from other yeast strains, and murine ASCA IgM can be experimentally induced with *Candida albicans*. *Inflamm Bowel Dis* 13: 1339-1346

Sendid B, Cornu M, Cordier C, Bouckaert J, Colombel JF, Poulain D (2024) From ASCA breakthrough in Crohn's disease and *Candida albicans* research to thirty years of investigations about their meaning in human health. *Autoimmun Rev* 23: 103486

Shirliff ME, Peters BM, Jabra-Rizk MA (2009) Cross-kingdom interactions: *Candida albicans* and bacteria. *FEMS Microbiol Lett* 299: 1-8

Taylor HB, Vasu C (2021) Impact of Prebiotic beta-glucan Treatment at Juvenile Age on the Gut Microbiota Composition and the Eventual Type 1 Diabetes Onset in Non-obese Diabetic Mice. *Front Nutr* 8: 769341

Taynton T, Barlow G, Allsup D (2022) PRO: Biomarker surveillance for invasive fungal infections without antifungal prophylaxis could safely reduce antifungal use in acute leukaemia. *JAC Antimicrob Resist* 4: dlac074

Underhill DM, Iliev ID (2014) The mycobiota: interactions between commensal fungi and the host immune system. *Nat Rev Immunol* 14: 405-416

Vincent J, Mignot G, Chalmin F, Ladoire S, Bruchard M, Chevriaux A, Martin F, Apetoh L, Rebe C, Ghiringhelli F (2010) 5-Fluorouracil selectively kills tumor-associated myeloid-derived suppressor cells resulting in enhanced T cell-dependent antitumor immunity. *Cancer Res* 70: 3052-3061

Whalley T, Dolton G, Brown PE, Wall A, Wooldridge L, van den Berg H, Fuller A, Hopkins JR, Crowther MD, Attaf M *et al* (2020) GPU-Accelerated Discovery of Pathogen-Derived Molecular Mimics of a T-Cell Insulin Epitope. *Front Immunol* 11: 296

Wheeler ML, Limon JJ, Bar AS, Leal CA, Gargus M, Tang J, Brown J, Funari VA, Wang HL, Crother TR *et al* (2016) Immunological Consequences of Intestinal Fungal Dysbiosis. *Cell Host Microbe* 19: 865-873

WHO, 2020. WHO fungal priority pathogens list to guide research, development and public health action.

2nd Jun 2025

Dear Prof. Shinohara,

We are pleased to inform you that your manuscript is accepted for publication and is now being sent to our publisher to be included in the next available issue of EMBO Molecular Medicine.

Your manuscript will be processed for publication by EMBO Press. It will be copy edited and you will receive page proofs prior to publication. You will soon be contacted by Springer Nature to sign your publishing license. When you login to the customer service website, please use the following token to waive the article publication charges. Should you experience any difficulty, please email publishing@embo.org.

Waiver token: *removed*
